# Mixture of neural fields for heterogeneous reconstruction in cryo-EM

**Axel Levy**[*]
Stanford University
axlevy@stanford.edu

**Rishwanth Raghu**[*]
Princeton University
rraghu@princeton.edu

**David Shustin**[*]
Princeton University
dshustin@princeton.edu

**Adele Rui-Yang Peng**
Princeton University
adelep@princeton.edu

**Huan Li**
Columbia University
hl3170@columbia.edu

**Oliver Biggs Clarke**
Columbia University
oc2188@cumc.columbia.edu

**Gordon Wetzstein**
Stanford University
gordon.wetzstein@stanford.edu

**Ellen D. Zhong**
Princeton University
zhonge@princeton.edu

## Abstract

Cryo-electron microscopy (cryo-EM) is an experimental technique for protein structure determination that images an ensemble of macromolecules in near-physiological contexts. While recent advances enable the reconstruction of dynamic conformations of a single biomolecular complex, current methods do not adequately model samples with mixed conformational and compositional heterogeneity. In particular, datasets containing mixtures of multiple proteins require the joint inference of structure, pose, compositional class, and conformational states for 3D reconstruction. Here, we present Hydra, an approach that models both conformational and compositional heterogeneity fully *ab initio* by parameterizing structures as arising from one of $K$ neural fields. We employ a new likelihood-based loss function and demonstrate the effectiveness of our approach on synthetic datasets composed of mixtures of proteins with large degrees of conformational variability. We additionally demonstrate Hydra on an experimental dataset of a cellular lysate containing a mixture of different protein complexes. Hydra expands the expressivity of heterogeneous reconstruction methods and thus broadens the scope of cryo-EM to increasingly complex samples. Webpage: https://hydra.cs.princeton.edu

## 1 Introduction

Structural information is key to understanding the function of macromolecular complexes, making protein structure determination a crucial tool in basic structural biology and rational drug design. Among experimental structure determination methods, cryogenic electron microscopy (cryo-EM) is unique in its capability to reveal dynamic information about large macromolecular complexes in near-native states.

---

[*]Equal contribution

38th Conference on Neural Information Processing Systems (NeurIPS 2024).

In single particle cryo-EM, a set of biomolecular complexes (i.e. particles) is flash-frozen and imaged with a transmission electron microscope. Each collected image consists of a noisy, randomly oriented projection of an individual particle of unknown identity or composition, frozen in an unknown state. Reconstruction algorithms processing these data without information from upstream algorithms are called *ab initio heterogeneous* reconstruction methods. Classical reconstruction techniques use a Bayesian approach to optimize a finite number of voxel-based representations [50, 45]. These algorithms enabled biologists to process datasets made of a mixture of different protein compositions and, thus illuminate the molecular details of fundamental biological processes. These methods, however, tend to aggregate all the proteins of the same composition in a *single class*, despite these proteins being trapped in different conformational states, due to thermal fluctuations prior to the freezing step. Another line of work has extensively studied the possibility of reconstructing continuous motion from cryo-EM datasets, using linear combinations of voxel arrays [43], neural-based representations [68, 69, 25, 26], Gaussian mixture models [5, 20], or combining a voxel array with a flow field [44]. These methods sometimes leverage a structural prior about proteins (e.g., proteins are made of a fixed number individual atoms) and enable the reconstruction of molecular motion at high resolution. However, none of these methods can handle datasets containing different types of proteins, thereby strongly limiting their application, especially in the context of *in situ* cryo-EM.

As of today, there exist no approaches to simultaneously reveal compositional (discrete) and conformational (continuous) heterogeneity in cryo-EM datasets, potentially limiting the structures that can be revealed from the data. Unraveling this information poses a nontrivial problem that sequential strategies cannot solve. Due to their low signal-to-noise-ratio, cryo-EM images cannot be clustered depending on the type of protein they contain prior to the reconstruction. Because orientations are unknown, the problem cannot be solved by handling compositional heterogeneity before conformational heterogeneity as consensus poses (i.e., orientations) are inaccurate for large motions.

Here, we introduce Hydra, a neural-based method for *ab initio* heterogeneous reconstruction in cryo-EM. Inspired by the success of implicit neural representations in cryo-EM, we extend the neural field representation of DRGN-AI [26] with a mixture model of $K$ neural fields. Using a new likelihood-based loss function, we simultaneously optimize orientations, conformations and class assignments and circumvent the pitfalls of sequential approaches. We demonstrate that our method allows neural-based methods to handle strong compositional heterogeneity and enables the simultaneous reconstruction of compositional and conformational heterogeneity with state-of-the-art accuracy. In an experimental cryo-EM dataset of a cell lysate mixture, we reveal three compositional states in a single pass, fully *ab initio*. We therefore make the following contributions:

- We develop a mixture of neural fields model for *ab initio* heterogeneous reconstruction in cryo-EM;
- We demonstrate that our method improves the expressiveness of neural-based methods for handling strong compositional heterogeneity;
- We enable simultaneous reconstruction of conformational and compositional heterogeneity beyond the limits of existing methods;
- We demonstrate the reconstruction of multiple protein complexes from an experimental dataset of an unpurified sample.

## 2 Related Work

### 2.1 Heterogeneous Reconstruction in Cryo-EM

**Discrete Variability.** Cryogenic electron microscopy offers the potential to reveal the structure of macromolecules in heterogeneous samples, where multiple types or multiple conformations are mixed together [24]. The first reconstruction algorithms handling 3D variability modeled the set of particles as a finite set of static structures. RELION [50] popularized the Bayesian approach to tackle heterogeneous reconstruction and the later introduced *multi-body refinement* tool [34] offered the possibility to segment static density maps and model continuous motion as a combination of rigid transformations. The software suite was recently improved with the Blush regularization tool [22, 4], leveraging a data-driven prior to enable the reconstruction of small protein-nucleic acid complexes ($\leq 40$ kDa). CryoSPARC [45] accelerated the inference with stochastic gradient descent and, in the

3DVA [43] extension, modeled continuous motion with a linear combination of density maps. These methods represent the state of the art for cryo-EM reconstruction but do not have the capability to represent complex non-linear motion and tackle the discrete heterogeneity (e.g., with "multi-class *ab initio*") before the continuous variability (e.g., with 3DVA or [21]), leading to inaccurate pose estimation for states exhibiting large motion.

**Non-Linear Methods for Heterogeneous Reconstruction.** In the past years, significant progress has been made in revealing non-linear dynamics from cryo-EM data. Manifold embedding [11] and Laplacian methods [33] are among the first attempts to model non-linear continuous motion but have only been applied to a small number of macromolecular complexes [8, 7]. HEMNMA [20] used a decomposition over the low-energy normal modes of an atomic model, thereby leveraging a prior over the structure and the dynamics of macromolecules. Herreros et al. [17] proposed the use of 3D Zernike polynomials and eliminated the need for pseudo-atomic models. CryoDRGN [69] used neural networks to continuously represent deformable density maps as well a variational auto-encoding framework to estimate conformational states. E2GMM [5], cryoFold [71], and DynaMight [51] followed this encoder-decoder framework and used Gaussian mixture models to represent density maps, thereby reducing the memory footprint of previous non-linear methods. 3DFlex [44] introduced the use of a parametric flow field to smoothly deform canonical 3D density maps, leveraging the knowledge that energetically favorable deformations tend to preserve the local geometry of proteins. Although these methods demonstrated the ability to reconstruct molecular motions and heterogeneity, they all need a coarse initialization of the density map, or the poses to be provided by an upstream reconstruction algorithm. In practice, this *ab initio* step is error prone in the presence of large conformational motions.

*Ab Initio* **Heterogeneous Reconstruction.** Recent works investigated the problem of reconstructing an ensemble of density maps where poses are unknown. The preliminary cryoDRGN method [68] tackled the *ab initio* reconstruction problem by combining traditional pose search algorithms with the encoder-decoder neural-based architecture, which was refined in cryoDRGN2 [70]. Multi-CryoGAN [15] sidestepped pose estimation by casting the reconstruction problem as a distribution matching problem and successfully revealed 3D variability in synthetic cryo-EM datasets. Rosenbaum et al. [49] showed that the auto-encoding framework could be applied to jointly estimate poses and conformations of atomic models, and Levy et al. [25] demonstrated a fully autoencoding framework for *ab initio* heterogeneous reconstruction on real benchmark datasets. DRGN-AI [26] recently introduced a hybrid pose search strategy combined with an autodecoding architecture to handle low-signal datasets and demonstrated *ab initio* heterogeneous reconstruction on challenging datasets containing a significant number of junk images. These methods extend neural-based reconstruction to scenarios where poses cannot be reliably estimated by any upstream algorithms, but have a limited capability to represent mixtures of biomolecular complexes, due to the limited capacity of a single neural representation.

Here, we propose to represent the landscape of accessible structures via an ensemble of $K$ neural networks, each *specialized* in representing the variability within a single compositional state. By jointly estimating structures and orientations, our method handles datasets that exhibit strong compositional heterogeneity and large motions.

## 2.2 Neural Fields for Large and Dynamic Scenes

**Dynamic Scenes.** In graphics, neural networks have also been used as light-weight, differentiable representations of continuously defined signals (e.g., occupancy fields [31], signed distance functions [37, 6], surface light fields [66], latent representation of appearance [52, 53], radiance fields [32, 1], and light fields [54]). In order to handle time-dependency and represent dynamic scenes, two approaches have been explored, as described in [39]. Similar to the cryo-EM method 3DFlex [44], *deformation-based* approaches apply a spatially-varying deformation to some canonical radiance field [38, 42, 59]. Methods following this strategy recover detailed dynamic scenes but are unable to model any topological variations. Similar to cryoDRGN [69], *modulation-based* approaches directly condition the radiance field of the scene on some latent vectors encoding temporal or dynamic information [13, 27, 65, 12]. These techniques are capable of modeling arbitrary deformations, topological changes, and other complex phenomena, but require additional regularization strategies

to avoid trivial, non-plausible solutions. HyperNeRF [39] was introduced as a combination of both approaches and enabled photo-realistic reconstruction of dynamic scenes with topological changes.

**Large-Scale Scenes.**   Due to their limited capacity, neural networks have essentially been used to represent single objects or small-scale scenes. Several methods [46, 47] have looked into the possibility of using an ensemble of neural fields to represent large scenes such as a neighborhood in the city of San Francisco [57]. Another approach is to provide extra capacity with a coarse 3D grid of latent codes [27, 55, 40] or a block-coordinate multi-scale decomposition [28]. These works focus on static reconstruction, and if dynamic objects are present in the scene, these are simply masked out [57]. Ost et al. [36] enabled the representation of complex, dynamic multi-object scenes by decomposing them into their static and dynamic parts and learning one neural radiance field per dynamic object.

Here, we use the *modulation-based* approach to handle the continuous motion of proteins and an ensemble of neural fields to increase the representational capacity of our model, allowing it to reconstruct compositional mixtures of proteins. In contrast with the works cited above, we need to estimate which compositional state each image belongs to – which is done using a variational approach – while simultaneously optimizing the ensemble of neural networks and individual image poses.

### 2.3   Adaptive Mixtures of Experts

A mixture of experts (MoE) model uses an ensemble of neural networks to process a given input [19]. Doing so, the input space can be partitioned into subspaces on which only one of the neural networks needs to become "expert". The mechanism with which the input space gets partitioned is usually referred to as the *gating* mechanism. Masoudnia and Ebrahimpour [29] partitions MoE approaches into two groups, depending on the gating mechanism and both how and when this mechanism is involved: (1) mixtures of explicitly localized experts (MELE) cluster the input data before the experts' training phase starts while (2) mixtures of implicitly localized experts (MILE) jointly optimize the expert networks and the gating mechanism. In cryo-EM, clustering methods operating directly on images are usually ineffective because of the high level of noise and because of the presence of other nuisance variables (orientation and conformation), making the MELE approach irrelevant.

Jacobs et al. [19] examined the use of different error functions (or loss functions) to optimize the expert networks and the gating mechanism, and the best performance was obtained using the negative log-probability of a Gaussian mixture model [35]. Several architectures have been explored, and one of the most applied is the mixture of MLP-experts (MME) [63, 9], where multi-layer perceptrons (MLPs) are used for both the experts and the gating networks. Our approach can be viewed as an application of the MILE method where, as in Jacobs et al. [19], the error function corresponds to the negative log-probability of a Gaussian mixture model. Due to the high level of noise, the gating mechanism ignores the content of input images and relies on an autodecoding framework.

## 3   Methods

In this section, we first describe the image formation model in single particle cryo-EM (3.1). We then define a 3D variability model that models both compositional and conformational heterogeneity (3.2). Based on these two first sections, we describe our system with a latent variable model (3.3) and explain our optimization method (3.4). The method is schematically described in Figure 1.

### 3.1   Image Formation Model

In cryo-EM, a purified solution of macromolecules is flash-frozen inside a thin layer of vitreous ice. Each molecule gets trapped in a random orientation with respect to the microscope and in a random conformational state, approximately following the Boltzmann distribution at ambient temperature [2]. The sample is exposed to an electron beam and individual 2D projections are extracted from micrographs via a step known as "particle picking". The reconstruction task therefore starts with a given set of $N$ images (or particles). Each image $I_i$ can be modeled as [60, 50]

$$I_i = C_i * \mathcal{P}_{\phi_i} V_i + \eta_i, \tag{1}$$

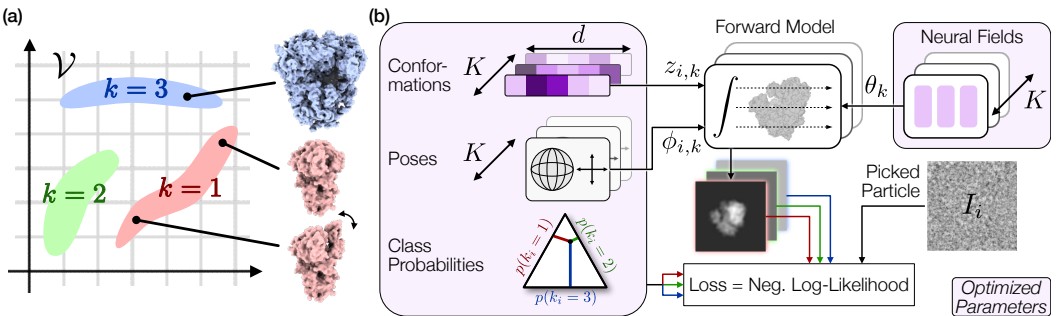

**Figure 1: Overview of Hydra. (a)** Schematic representation of the space of energetically plausible density maps in a heterogeneous cryo-EM dataset. We approximate this space with a finite union of low-dimensional manifolds. The compositional states (or classes) are labeled by $k$. The "conformation" within class $k$ refers to intrinsic coordinates within the $k$-th manifold. **(b)** Optimization pipeline. The conformations, poses, class probabilities and neural fields are optimized such as to maximize the likelihood of the observed images ("picked particles") under the model described in Section 3.3.

where $C_i$ models the the Contrast Transfer Function (CTF), $V_i$ is a scalar 3D field ($\mathbb{R}^3 \to \mathbb{R}$) known as the "electron scattering potential" or the "density map", $\phi_i = (\mathbf{R}_i, \mathbf{t}_i)$ is a "pose" ($\mathbf{R}_i \in \mathrm{SO}(3)$, $\mathbf{t}_i \in \mathbb{R}^2$) and $\mathcal{P}$ projects $V_i$ on a 2D grid:

$$\mathcal{P}_{(\mathbf{R},\mathbf{t})}V_i = \left\{ \int_t V\left(\mathbf{R} \cdot [x_{m,n} - t_x, y_{m,n} - t_y, t]^T\right), (m,n) \in \{1,...,D\}^2 \right\}. \quad (2)$$

$\eta_i$ models isotropic Gaussian noise ($\eta_i \sim \mathcal{N}(0, \sigma^2)$). In a typical experiment, $N$ can vary between $10^5$ and $10^7$; The signal-to-noise ratio can vary between $10^{-1}$ and $10^{-2}$.

## 3.2 Heterogeneity Model

Structural heterogeneity among the macromolecules can originate (1) from continuous motion along a small number of *degrees of freedom*, or (2) from discrete compositional changes. We refer to the first kind of heterogeneity as "conformational heterogeneity" and to the second one as "compositional heterogeneity".

To model this mathematically, we make the assumption that all the density maps $V_i$ belong to a finite union of low-dimensional manifolds:

$$\forall i \in \{1, \dots, N\}, V_i \in \{\mathcal{V}(z; \theta_k), z \in \mathbb{R}^d, k \in \{1, \dots, K\}\}, \quad (3)$$

where $K \in \mathbb{N}$, $d \in \mathbb{N}$ and, for all $k$, $\theta_k \in \Theta$ is an unknown parameter. In other words, we assume that there exist $K$ compositional states and that the conformational motion of state $k$ can essentially be described with $d$ degrees of freedom.

## 3.3 Latent Variable Model

We statistically describe the set of observed images with a latent variable model. Here, the *latent variables* are the poses $\phi_i$, the conformations $z_i$ and the class identities $k_i$, while $\{\theta_k\}_{k=1}^K$ is a set of *shared parameters*.

Given the image formation model described by Eq. 1 and the heterogeneity model described by Eq. 3, each observed image can be seen as a sample from a multivariate mixture model:

$$p(I_i) = \sum_{k=1}^K p(k_i = k) \iint p(\phi|k_i = k)p(z|k_i = k)\mathcal{N}(C_i * \mathcal{P}_\phi \mathcal{V}(z; \theta_k), \sigma^2)\mathrm{d}\phi\mathrm{d}z. \quad (4)$$

We parameterize a probability distribution over the class identity, such that:

$$\forall i \in \{1, \dots, N\}, \forall k \in \{1, \dots, K\}, \quad p(k_i = k) = \mathrm{softmax}(\mathbf{s}_i)_k \doteq \frac{\exp(s_{i,k})}{\sum_j \exp(s_{i,j})}, \quad (5)$$

where $\mathbf{s}_i \in \mathbb{R}^K$ is a free parameter called the "score". For each $k$, we parameterize point estimates of the continuous latent variables:

$$
\begin{aligned}
p(\phi|k_i = k) &= \delta(\phi - \phi_{i,k}), \quad \phi_{i,k} \in \text{SO}(3) \times \mathbb{R}^2 \\
p(z|k_i = k) &= \delta(z - z_{i,k}), \qquad\quad z_{i,k} \in \mathbb{R}^d.
\end{aligned}
\tag{6}
$$

Under the model in Equation (4) and the variational parameterization described above, the negative log-likelihood of an image is given by

$$
\begin{aligned}
&\ell_i(\{\phi_{i,k}\}_{k=1}^K, \{z_{i,k}\}_{k=1}^K, \mathbf{s}_i; \{\theta_k\}_{k=1}^K) \\
&\quad = -\log \sum_{k=1}^K \exp\left( -\frac{1}{2\sigma^2} \|I_i - C_i * \mathcal{P}_{\phi_{i,k}} \mathcal{V}(z_{i,k}; \theta_k)\|_2^2 + \log(\text{softmax}(\mathbf{s}_i)_k) \right)
\end{aligned}
\tag{7}
$$

In our implementation, each low-dimensional manifold $\mathcal{V}(.; \theta_k)$ is implemented with a residual MLP. We provide further details on the architecture of the neural networks in the supplementary materials.

### 3.4   Hybrid Optimization Strategy

We aim at minimizing the negative log-likelihood of the set of observed images,

$$
\mathcal{L} = \sum_{i=1}^N \ell_i(\{\phi_{i,k}\}_{k=1}^K, \{z_{i,k}\}_{k=1}^K, \mathbf{s}_i; \{\theta_k\}_{k=1}^K),
\tag{8}
$$

over $(\{\phi_{i,k}\}, \{z_{i,k}\}, \{\mathbf{s}_i\}, \{\theta_k\})$. All the parameters are optimized using a combination of gradient-based optimization and exhaustive search. Here, we use an autodecoding framework and do not *amortize* the inference of latent variables with an encoder, following the observation in Levy et al. [26] that encoders tend to memorize images in highly-noisy setups.

We handle the minimization over $\{z_{i,k}\}$, $\{\mathbf{s}_i\}$ and $\{\theta_k\}$ with stochastic gradient descent. The minimization over the poses $\{\phi_{i,k}\}$ is more challenging, due to the existence of local minima. We take inspiration from the two-step pose estimation strategy introduced in DRGN-AI [26] and adapt it to cope with the simultaneous representation of multiple maps. First, optimization over poses is done with hierarchical pose search (HPS) in alternation with stochastic gradient descent on the other variables. On a given random *minibatch* of indices $\mathcal{I} \subset \{1, \ldots, N\}$,

$$
\forall i \in \mathcal{I}, \forall k \in \{1, \ldots, K\}, \quad \phi_{i,k} \leftarrow \underset{\phi \in \Omega}{\arg\min} \|I_i - C_i * \mathcal{P}_\phi \mathcal{V}(z_{i,k}; \theta_k)\|_2^2,
\tag{9}
$$

where $\Omega$ is a predefined grid in the space of poses (SO(3) $\times \mathbb{R}^2$). See [26] and the supplementary for more details on how the minimization can be done efficiently with a hierarchical strategy. This robust but computationally-expensive strategy is used for a predefined number of steps, usually between $5 \times 10^5$ and $2 \times 10^6$. After that, most poses are located in the basin of attraction of their global optimum and switching to stochastic gradient descent (SGD) makes computation more efficient while improving pose accuracy (poses are not constrained to belong to a predefined grid of fixed resolution during SGD).

At the end of optimization, the estimated class of image $I_i$ is simply given by the index $k_i$ of the largest entry in $\mathbf{s}_i \in \mathbb{R}^K$.

## 4   Experiments

We run three experiments to evaluate Hydra. In Section 4.1, we show that our method improves the expressiveness of neural-based methods and can reveal strong compositional heterogeneity fully *ab initio*. In Section 4.2, we use Hydra to reveal compositional heterogeneity in an experimental dataset containing protein complexes of diverse sizes, in a single run. In Section 4.3, we demonstrate the simultaneous reconstruction of compositional and conformational heterogeneity.

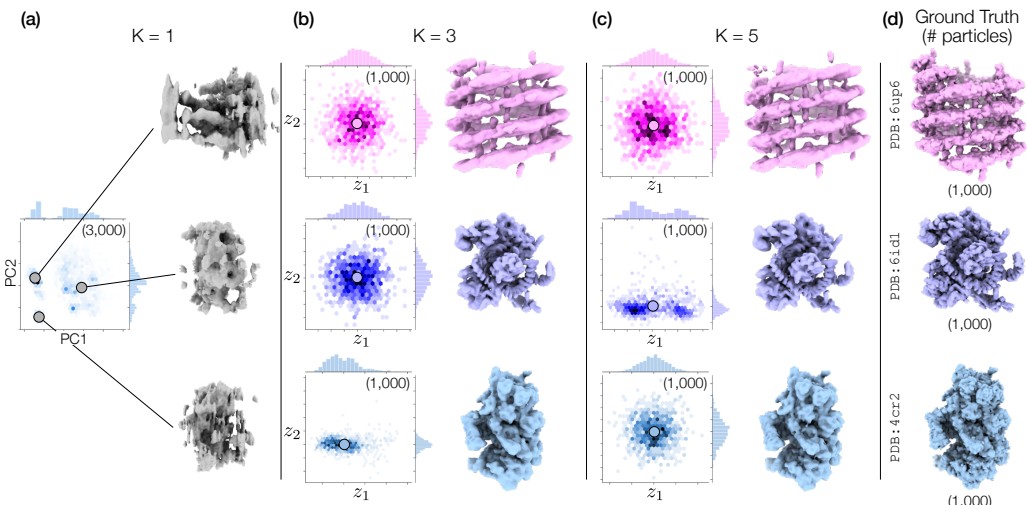

**Figure 2: Hydra captures strong compositional heterogeneity in the `tomotwin3` dataset. (a-c)** Reconstructed densities and estimated conformations with $K \in \{1, 3, 5\}$. We report the number of particles in each class between parenthesis. We represent density maps using isosurfaces. **(a)** With $K = 1$ (DRGN-AI), the model fails to reconstruct the three density maps, in spite of using $d = 8$ dimensions to represent conformations. **(b)** With $K = 3$ ($d = 2$), Hydra recovers the three density maps with perfect classification accuracy. **(c)** With $K = 5$ ($d = 2$), the model is over-parameterized and 2 classes out of 5 end up empty at the end of optimization. **(d)** Ground truth density maps for the `tomotwin3` dataset.

| | **ARI ↑** | **Per-image FSC ↑** | | |
|---|---|---|---|---|
| **Model** | All | PDB `6up6` | PDB `6id1` | PDB `4cr2` ↑ |
| Hydra ($K = 3$) | **1.00** | $0.25 \pm 0.03$ | $\underline{0.394 \pm 0.001}$ | $\mathbf{0.367 \pm 0.001}$ |
| Hydra ($K = 5$) | **1.00** | $\underline{0.25 \pm 0.03}$ | $\mathbf{0.396 \pm 0.001}$ | $\mathbf{0.367 \pm 0.001}$ |
| DRGN-AI ($K = 1$) | 0.59 | $0.0242 \pm 0.003$ | $0.040 \pm 0.005$ | $0.042 \pm 0.004$ |
| CryoDRGN2 | 0.36 | $0.08 \pm 0.01$ | $0.070 \pm 0.006$ | $0.3 \pm 0.1$ |
| CryoSPARC | **1.00** | **0.284** | 0.367 | 0.338 |

**Table 1: Hydra captures compositional heterogeneity in a challenging synthetic dataset and outperforms other neural-based methods.** The classification accuracy is evaluated for each method using the adjusted Rand index (ARI) [18]. To evaluate each method's reconstruction quality, we use the mean area under the Fourier shell correlation (FSC) curve for 20 images per class (we report $\pm 1$ standard deviation). We **bold** the best result, and underline the second best result.

## 4.1 *Ab initio* reconstruction of compositional heterogeneity

We evaluate Hydra on `tomotwin3`, a synthetic dataset of 3,000 images emulating a protein sample containing multiple species with static structures. We selected the 6th, 7th, and 8th largest proteins by atomic weight from the TomoTwin training dataset [48], which correspond to entries `6up6`, `6id1`, and `4cr2` in the RCSB PDB (Protein Data Bank) [3]. We rendered 1,000 synthetic images of each specimen following the cryo-EM image formation model. For each protein structure, we simulated density maps with the ChimeraX `molmap` command [30, 58]. We padded the density maps to a box size of $D = 384$ pixels and centered them. We sampled 1,000 orientations per ground truth class uniformly from $SO(3)$. We projected the density maps using Eq. 1 and applied a translation vector uniformly chosen from a 30-pixel-wide box centered at the origin. We applied a CTF sampled from EMPIAR-11247 [10] and added Gaussian noise (SNR = 0.01). We downsampled each image to a box size of $D = 128$, yielding 3,000 images in total (samples shown in Figure S1).

We compare the performance of Hydra to several state-of-the-art *ab initio* methods for resolving compositional heterogeneity in cryo-EM datasets, including DRGN-AI [26], cryoDRGN2 [70], and cryoSPARC [45]. Hydra only uses 2 dimensions for the conformational space, while DRGN-AI and CryoDRGN2 are evaluated with a more expressive 8-dimensional space. We test Hydra using the correct number of classes ($K = 3$, the true number of specimens present in the sample) and

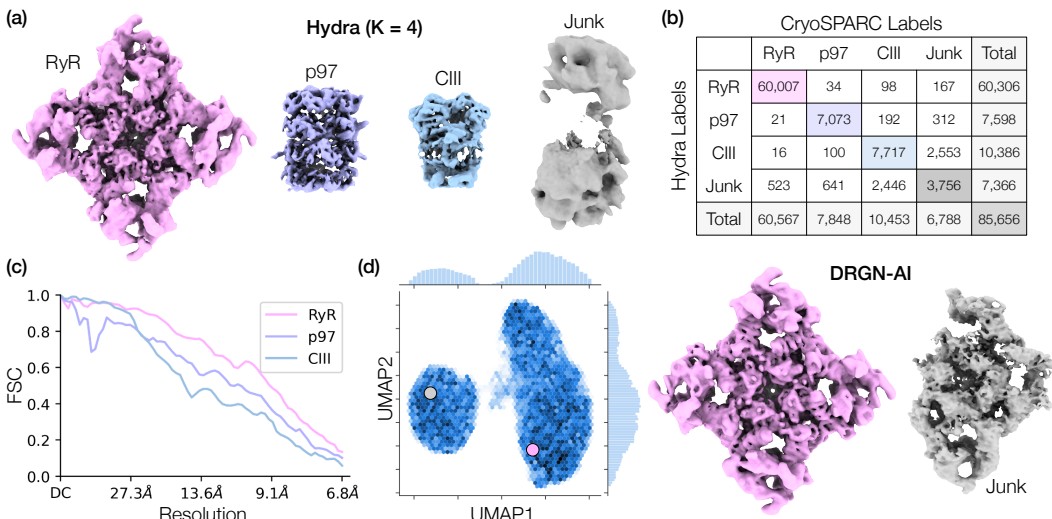

**Figure 3: Hydra captures compositional heterogeneity in a real dataset containing a mixture of membrane and soluble protein complexes.** **(a)** Density maps obtained with Hydra ($K = 4$) on the Ryanodine receptor dataset. **(b)** Confusion matrix between Hydra and cryoSPARC $K = 6$ heterogeneous refinement (three classes representing RyR were combined for analysis). **(c)** Fourier shell correlation (FSC) between the Hydra density maps and refined cryoSPARC density maps. **(d)** *Left:* latent space plot and *right:* representative density maps from each of the latent space clusters from DRGN-AI.

using an over-parameterized setup ($K = 5$ in Figure 2 and $K = 7$ in Figure S2). For DRGN-AI and CryoDRGN2, we classify each image using $k$-means clustering on the conformational space with 3 clusters. We evaluate per-image FSC for each class using the dataset's ground truth class labels.

In Table 1, we report results for each configuration of Hydra and other methods, (best of three replicas with distinct seeds for Hydra and DRGN-AI, full results in Table 3). We observe that other methods with implicit neural volume representations, including DRGN-AI and CryoDRGN2, fail to capture the compositional heterogeneity of the dataset. Hydra matches the classification quality of cryoSPARC, which models independent, discrete mixtures. Hydra also outperforms all methods on per-image FSC, a distributional volume reconstruction quality metric based on the Fourier Shell Correlation (FSC) [68]. We provide additional results with a larger version of this dataset in Figure S3 and additional metrics (including roto-translation accuracy) in Table 4.

## 4.2 *Ab initio* reconstruction of an experimental cryo-EM mixture dataset

We evaluate our method on an experimental dataset of a protein mixture from red blood cell lysate subjected to density-gradient centrifugation followed by chemical cross-linking. Through an exhaustive, expert-driven data processing pipeline in cryoSPARC [45], we determined that the dataset consisted of substantial compositional heterogeneity from three main component protein complexes: the membrane proteins ryanodine receptor (RyR) and mitochondrial respiratory chain complex III (CIII), and a dimeric complex of the soluble valosin-containing protein (p97). A sweep of $K$-values from cryoSPARC ab-initio reveals that $K \geq 5$ is necessary in order for cryoSPARC to reconstitute distinct RyR, p97, CIII, and junk classes (Figure S4).

In Figure 3, we compare the performance of DRGN-AI [26] with Hydra. Figures 3d and S5 show that DRGN-AI successfully partitions the dataset into two clusters corresponding to the RyR and non-RyR particles; however, DRGN-AI is unable to learn distinct shapes for all three discrete structures and instead appears to learn structural artifacts from the non-RyR particles that match the overall RyR shape. By contrast, Hydra with $K = 4$ successfully separates the particles into the three component protein complexes and a fourth junk class (Figures 3 and S6). As a baseline for comparison, we also carried out the typical workflow of first generating poses from a consensus reconstruction in cryoSPARC, followed by fixed-pose DRGN-AI to separate the heterogeneity. As can be seen in Figure S7, DRGN-AI recovers a high resolution RyR density but fails to learn non-RyR protein structures when training from poses from homogeneous reconstruction. A multi-step reconstruction with DRGN-AI does not either reveal the CIII class (Figure S8).

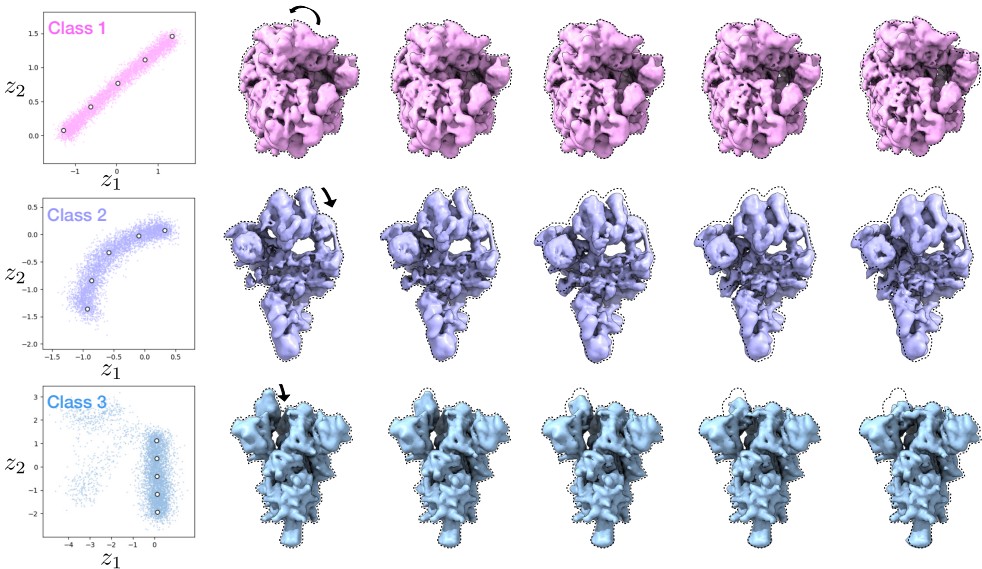

**Figure 4: Hydra effectively recovers both compositional and conformational heterogeneity in the `ribosplike` dataset.** Particles within each latent space are colored by class. Representative density maps are generated from the latent points denoted in white dots.

### 4.3 *Ab initio* reconstruction of conformational and compositional heterogeneity.

We prepare a synthetic dataset exhibiting both compositional and conformational heterogeneity (`ribosplike`). For each of the pre-catalytic spliceosome, 80S ribosome, and SARS-CoV-2 spike protein, 5,000 images are generated from different conformational states of the macromolecule along a 1-dimensional trajectory of motion [41, 64, 61] yielding a dataset of 15,000 images (See SI Section F for additional details and Figure S1 for sample images).

As seen in Figure 4, our method with $K = 3$ nearly perfectly separates the particles into their three classes. Within each neural field, the conformational change of the dominant particle type is well captured both by the conformational space. Due to the trimeric structure of the spike protein, we observe three continuous manifolds in the conformational space, though the majority of spike particles are aligned to one of the symmetries. Tables 2 and 5 compare the performance of Hydra with other *ab initio* baselines, demonstrating superior performance in classification accuracy as measured by the Adjusted

| Model | Img-FSC ↑ | ARI ↑ | Pose Err. ↓ |
|---|---|---|---|
| **Hydra (K=3)** | **0.414** | **0.997** | **1.070** |
| DRGN-AI | 0.207 | 0.994 | 45.379 |
| CryoDRGN2 | 0.399 | 0.986 | 1.2120 |
| CryoSPARC | 0.344 | 0.972 | 1.576 |

**Table 2: Hydra outperforms state-of-the-art methods on the `ribosplike` dataset.** Quantitative results include reconstruction quality (per-image FSC), particle classification (ARI), and median pose accuracy (geodesic distance between rotations, in degrees).

Rand Index (ARI) and volume reconstruction quality (per-image FSC). It additionally achieves the lowest pose error amongst all methods. Representative reconstructions for baseline methods are shown in the supplementary material (Figure S9).

## 5 Discussion

This work introduces Hydra, a method that expands the expressiveness of neural-based models for heterogeneous reconstruction in cryo-EM. Hydra models structures as arising from 1 of $K$ neural fields and is designed to capture heterogeneity in datasets containing a mixture of multiple species. Notably, our method runs fully *ab initio*, i.e. does not rely on pre-estimated poses, which are not straightforward to obtain in cases of strong compositional heterogeneity. We evaluated Hydra on both synthetic and experimental datasets, showed improved performance over previous neural-based methods on compositionally heterogeneous datasets, and demonstrated that it could simultaneously handle compositional and conformational heterogeneity.

Our method can be seen as an instance of a mixture of experts model, where the gating mechanism is handled by an autodecoding framework, due to the low signal-to-noise ratio in the input data. However, predicting classes with a neural network can be viewed a *classification* task and may be easier than pose or conformation estimation. We view the possibility of using a (potentially pretrained) neural network for classification, thereby making use of amortized inference, as an exciting avenue for future work.

One important limitation of our approach is the need to specify the number of classes $K$ prior to reconstruction. Other methods that cope with compositional heterogeneity, like RELION [50] and cryoSPARC [45], possess the same requirement, and this is solved in practice by running the algorithm several times with a sweep on the hyperparameter $K$. As this process is time and energy-consuming (a single reconstruction can run for up to 4 GPU-days on high-end GPUs), more efficient methods for hyperparameter selection, for example, with an adaptive and/or hierarchical strategy, is an impactful direction for future work. We would also like to note that, in its current implementation, Hydra must give the same latent dimension ($d$) to all the classes, but this constraint could be relaxed with, for-example, a dictionary-based representation for the conformations.

With Hydra, we broaden the scope of datasets that cryo-EM reconstruction algorithms can process. Our method would be especially suitable for data collected *in situ* (i.e., inside the cell), where mixtures of dynamic biomolecular complexes coexist. This data is usually acquired by cryogenic electron tomography (cryo-ET), where the sample is progressively tilted throughout the data collection process. Extending our method to enable subtomogram averaging of cryo-ET data is another potential future direction that we hope can enable future discoveries in structural biology and facilitate the design of new therapeutic compounds.

## 6 Acknowledgments

The author(s) are pleased to acknowledge that the work reported on in this paper was substantially performed using the Princeton Research Computing resources at Princeton University which is a consortium of groups led by the Princeton Institute for Computational Science and Engineering (PICSciE) and Office of Information Technology's Research Computing. The Zhong lab gratefully acknowledges support from the Princeton Catalysis Initiative, Princeton School of Engineering and Applied Sciences, and Generate Biomedicines. The funders had no role in study design, data collection and analysis, decision to publish or preparation of the manuscript.

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

# Appendix

## A   Architectural Details

Each manifold of density maps $\mathcal{V}(,;\theta_k)$ is implemented as a neural network. Conditioned on a conformation $z \in \mathbb{R}^d$, $\mathcal{V}(,;\theta_k) : \mathbb{R}^3 \rightarrow \mathbb{R}$ represents the Hartley transform of the 3D electron scattering potential of a single particle. The frequency coordinate $\mathbf{k} \in [-0.5, 0.5]^3$ is expanded in a sinusoidal basis using Fourier features [56] (64 base frequencies are randomly sampled from a 3D Gaussian distribution of standard deviation 0.5). The neural network contains 3 hidden residual layers [16] of size 128 with ReLU nonlinearities, without any normalization schemes.

## B   Pose Estimation

"Hierarchical pose search" (HPS) is done on a predefined grid over $SO(3) \times \mathbb{R}^2$ (4,608 rotations and 49 translations in $[-10 \text{ pix.}, 10 \text{ pix.}]$). The first 8 poses minimizing the reprojection error are kept and refined with a local search over 8 neighbors during 4 additional steps. The images are band-limited during pose search and the cutoff frequency increases linearly from $k_{\min}$ to $k_{\max}$ ($k_{\min} = 6$, $k_{\max} = 16$ in (image length)$^{-1}$). Grids are parameterized using the Hopf fibration [67], product of the Healpix [14] grid on the 2-sphere and a regular grid on the circle.

Once 2,000,000 images have been fed to the model (with a minimum of 2 epochs of pose search), the pose estimation strategy switches to stochastic gradient descent (SGD). The poses $\phi_{i,k}$ are initialized with the last poses estimated by hierarchical pose search.

## C   Conformation Estimation

The conformations $\phi_{i,k}$ are independently optimized by SGD. They are initialized randomly from a $d$-dimensional Gaussian distribution of standard deviation 0.1. Unless stated otherwise, the dimension $d$ of the conformations is 2.

## D   Score Estimation

The scores $\mathbf{s}_i \in \mathbb{R}^K$ are optimized by SGD and converted into probability vectors of dimension $K$ with a softmax operator.

## E   Optimization Parameters

We use the Adam optimizer [23] without weight decay and with the following learning rates: 0.1 for the scores, 0.01 for the conformations, 0.001 for the poses, and 0.0001 for the weights of the neural networks.

## F   Synthetic Datasets

**Simulation of compositional heterogeneity.** We first describe the construction of `tomotwin3`, the synthetic dataset with compositional heterogeneity only. We selected the 6th, 7th, and 8th largest proteins by atomic weight from the training dataset used in TomoTwin [48], which correspond to entries `6up6`, `6id1`, and `4cr2` in the RCSB PDB (Protein Data Bank) [3]. We simulate density maps for each entry using the ChimeraX `molmap` command [30, 58] using a grid spacing of 1.5 Å/px and a resolution of 3.0 Å/px. We pad each density map to a box size of $D = 384$ pixels and center it. We sample 3,000 orientations uniformly from $SO(3)$, and 3,000 translation vectors in a 30-pixel-wide box around the origin (1,000 poses for each PDB entry). We project each density map by applying the corresponding orientation, projecting using Eq. 1, and applying the corresponding translation vector. We apply a contrast transfer function (CTF) uniformly sampled from EMPIAR-11247 [10], a representative cryo-EM dataset. We add Gaussian noise to each image to reach a signal-to-noise ratio (SNR) of 0.01. We downsample each image to an image size of $D = 128$ pixels.

We evaluated Hydra on `tomotwin3` with a 2-dimensional conformational space. We perform HPS on 100,000 images (33 epochs), followed by 100 epochs of SGD pose optimization. We set the batch size to 64 during SGD pose optimization. The score table learning rate is set to 0.01, and $\sigma$ is set to 0.1. We evaluate Hydra using both the correct number of classes ($K = 3$) and an overparametrized configuration ($K = 5$). All other parameters are set to default values.

We train CryoDRGN2 v3.3.0 for 30 epochs using an 8-dimensional latent space, an encoder width of 1024, 3 encoder layers, and a decoder width of 1024. We evaluate DRGN-AI using an 8-dimensional conformational space, 100k images (33 epochs) of HPS followed by 100 epochs of SGD pose estimation. cryoSPARC *ab initio* was run with $K = 3$ classes. All other parameters are set to default values.

When calculating volume metrics, we compare the reconstructed density map to a downsampled ground truth density map ($D = 128$ pixels). All experiments on `tomotwin3` are run on one NVIDIA A100 GPU. Our experiments required 4h00min for $K = 3$ and 6h20min for $K = 5$. DRGN-AI ran in 1h20min and cryoDRGN2 in 1h50min.

**Simulation of conformational and compositional heterogeneity.** For the pre-catalytic spliceosome, we obtain a trained cryoDRGN model from Zenodo[2] and generate 500 density maps of box size $D$=256 at equally spaced points along the first principal component of the latent space [41] [69]. For the 80S ribosome, we likewise obtain a trained cryoDRGN model from Zenodo and generate 500 density maps of box size $D$=256 at equally spaced points along a linear trajectory connecting the embeddings of particles 34570 and 60629 in the latent space [64, 69]. For the SARS-CoV-2 spike protein, a cryoDRGN model was trained on a filtered subset of the dataset of Walls et al. [62], for 25 epochs with a circular image mask of dimension 64, latent dimension of 8, and all other hyperparameters at default. Then, we generate 500 density maps of box size $D$=256 at equally spaced points along the first principal component of the latent space [61].

We next match the pixel sizes of the density maps by standardizing them to that of the spliceosome. In particular, we downsample the ribosome density maps to $D$=228 and pad back to $D$=256, and downsample the spike protein density maps to $D$=198 and pad back to $D$=256. Then, we generate and apply a soft mask for each density map, at thresholds of 0.03, 0.05, and 0.1 for the spliceosome, ribosome, and spike protein, respectively, and with 25Å of dilation and a 15Å cosine edge. Next, we normalize density map intensity values such that the signal ratios between different particle types match approximate "true" signal ratios. We calculate the true signal ratios by running the ChimeraX *molmap* command on the PDBs of each particle (spliceosome 5NRL, ribosome 3J79, spike 6VXX and 6VYB), summing the density within each density map, and calculating signal ratios (where the signal of the spike protein is taken to be the average over the two PDBs). Then, the intensities of the 500 ribosome density maps and 500 spike protein density maps are scaled such that the ratios of total intensities across all density maps for ribosome to spliceosome, and spike to spliceosome, match the true ratios.

Finally, we generate images from all the density maps. For each density map, 100 projection images are generated with uniformly sampled rotations from SO(3) and uniformly sampled in-plane translations within [-20, 20] pixels. CTF is applied to each image, with values drawn (with replacement) from the experimental CTF values of the spliceosome dataset [41]. Noise is added to all images at an SNR of 0.1, with the variance of the signal computed over all 15k particles. Lastly, the images are downsampled from $D$=256 to $D$=128.

We train Hydra for 100,000 images of HPS before 100 epochs of SGD. We train cryoDRGN2 for 90 epochs. All other hyperparameters for all methods are set to their defaults. For single-class DRGN-AI and cryoDRGN2, the predicted classes for Adjusted Rand Index calculation are obtained by $k$-means clustering the latent space with $k$=3. Image-FSC is computed over 50 images equally spaced along the true conformational trajectory, per each of the 3 particle types (150 images total). Experiments for Hydra were run using 2 NVIDIA V100 GPUs, and experiments for baselines were run using 1 NVIDIA V100 GPU.

---

[2] `https://zenodo.org/records/4355284`

# G   Real Dataset

**Dataset details.** 85,656 particles were manually picked from one round of 2D classification on cryoSPARC of a larger dataset of 148,596 particles. The particles were downsampled from a box size of $D = 300$ pixels and 1.66 Å/pixel to $D = 150$ pixels at 3.32 Å/pixel.

**DRGN-AI and Hydra.** All training for DRGN-AI and Hydra were carried out on 4 A100 NVIDIA GPUs with 80GB memory. For *ab initio* DRGN-AI, we ran DRGN-AI with default parameters (latent dimension $d = 8$), with 500k images for HPS followed by 100 epochs of SGD. We verified that single-class DRGN-AI could separate the RyR and non-RyR particles by selecting the particles belonging to the non-RyR cluster (as visualized in the latent space) for downstream single-class DRGN-AI analysis, and a second run of single-class DRGN-AI with default parameters, 500k images of HPS and 100 epochs SGD on the non-RyR particles recovered a p97 density map and a non-p97 cluster in the latent space (followed by a third round of DRGN-AI with the same parameters on the non-p97 particles). For Hydra, we trained the model with 2 million images for HPS followed by 100 epochs of SGD and latent dimension $d = 2$. After an initial sweep of the hyperparameters for learning rate and high-entropy prior $\sigma$ using $K = 3$ or $K = 4$, the optimal parameters for this dataset were determined to be a learning rate of 0.1 and $\sigma = 10.0$. In order to reduce the resource demand of Hydra, we lowered the hypervolume dimension to 128 and reduced the number of hidden layers in the hypervolume to 2, as well as decreased the SGD batch size to 64. For fixed-pose DRGN-AI with consensus cryoSPARC poses, poses from a cryoSPARC homogeneous refinement job were used as input for optimization by single-class DRGN-AI with default parameters.

**CryoSPARC processing.** We performed a sweep of $K$-values for 3D classification using cryoSPARC *ab initio* from $K = 1$ through $K = 8$ (all other parameters set to default values), and determined that $K < 5$ is insufficient to separate the junk particles from protein. $K = 6$ *ab initio* yielded the best separation of particles into the four different classes, with three different RyR density maps produced. We performed heterogeneous refinement on the best *ab initio* $K = 6$ replicate for comparison to Hydra. In accordance with best practices for data processing, we performed *ab initio* with the downsampled $D = 150$ particles but used undownsampled particles boxed at $D = 300$ to generate the most accurate poses during refinement against the low-resolution density maps generated by *ab initio*.

# H   Additional Files

We provide the following supplementary files:

- Three movies illustrating the continuous motion of the ribosome, spliceosome and spike, as shown in Figure 4. The movies were obtained by sampling 9 points on a linear trajectory of the latent space, for each compositional state.
- The 10 volumes shown in Figure 4 for the ribosome and the spliceosome, in `mrc` format.

# I Additional Figures

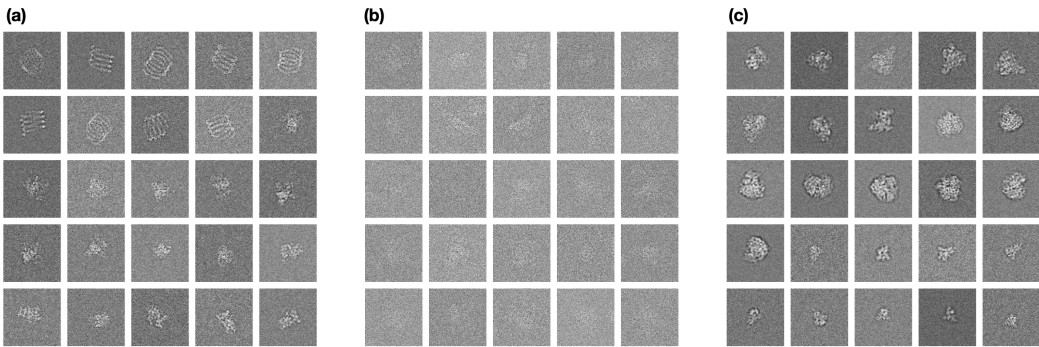

**Fig. S1: 25 representative sample images from each of the referenced three datasets. (a)** Sample images for the `tomotwin3` synthetic dataset, $D = 128$, 4.5 Å/pix. **(b)** Sample images for the experimental ryanodine receptor dataset, $D = 150$, 3.32 Å/pix. **(c)** Sample images for the `ribosplike` synthetic dataset, $D = 128$, 4.24 Å/pix.

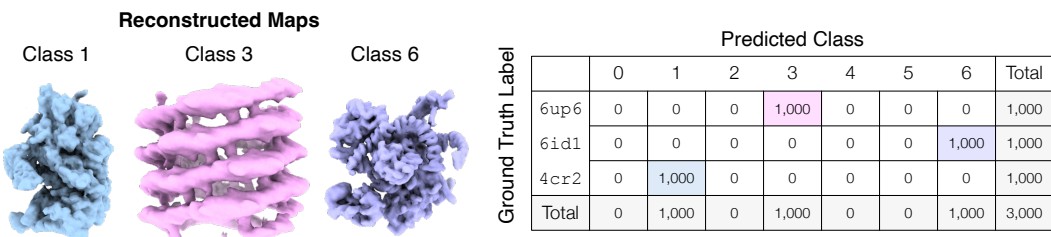

**Fig. S2:** Results on the `tomotwin` dataset (3k particles) with a larger-than-optimal value for $K$ ($K = 7$). Reconstructed maps on the left and confusion matrix on the right. Hydra is able to accurately reconstruct the three states in the dataset but four classes end up empty.

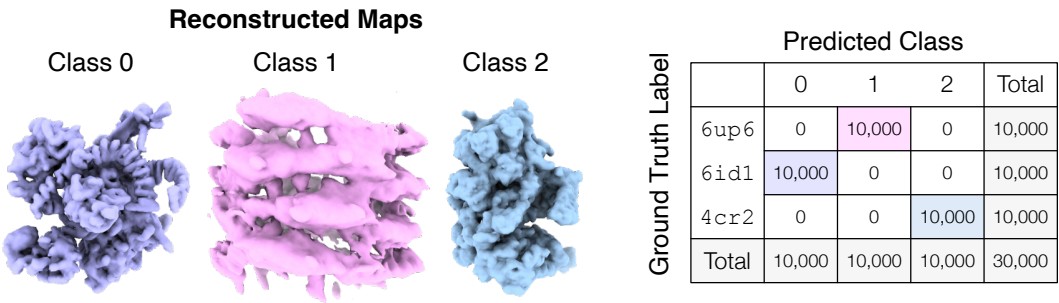

**Fig. S3:** Results on the `tomotwin` dataset (30k particles, $K = 3$).

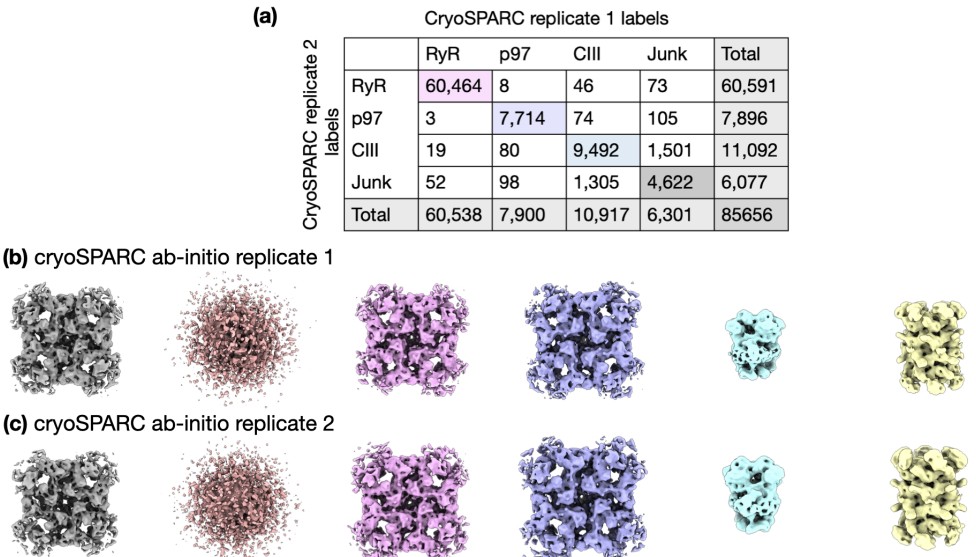

**(a)**

CryoSPARC replicate 1 labels

| CryoSPARC replicate 2 labels | | RyR | p97 | CIII | Junk | Total |
|---|---|---|---|---|---|---|
| | RyR | 60,464 | 8 | 46 | 73 | 60,591 |
| | p97 | 3 | 7,714 | 74 | 105 | 7,896 |
| | CIII | 19 | 80 | 9,492 | 1,501 | 11,092 |
| | Junk | 52 | 98 | 1,305 | 4,622 | 6,077 |
| | Total | 60,538 | 7,900 | 10,917 | 6,301 | 85656 |

**(b)** cryoSPARC ab-initio replicate 1

**(c)** cryoSPARC ab-initio replicate 2

**Fig. S4: cryoSPARC *ab initio* $K = 6$ shows comparable levels of classification uncertainty as Hydra $K = 4$, with cryoSPARC *ab initio* showing significant classification uncertainty between particles from the CIII and junk classes, similar to the comparison between Hydra class assignments and cryoSPARC *ab initio* reference labels. (a)** Confusion matrix between two $K = 6$ replicates of cryoSPARC *ab initio*. **(b)** and **(c)**: reconstructed density maps from the two cryoSPARC *ab initio* replicates.

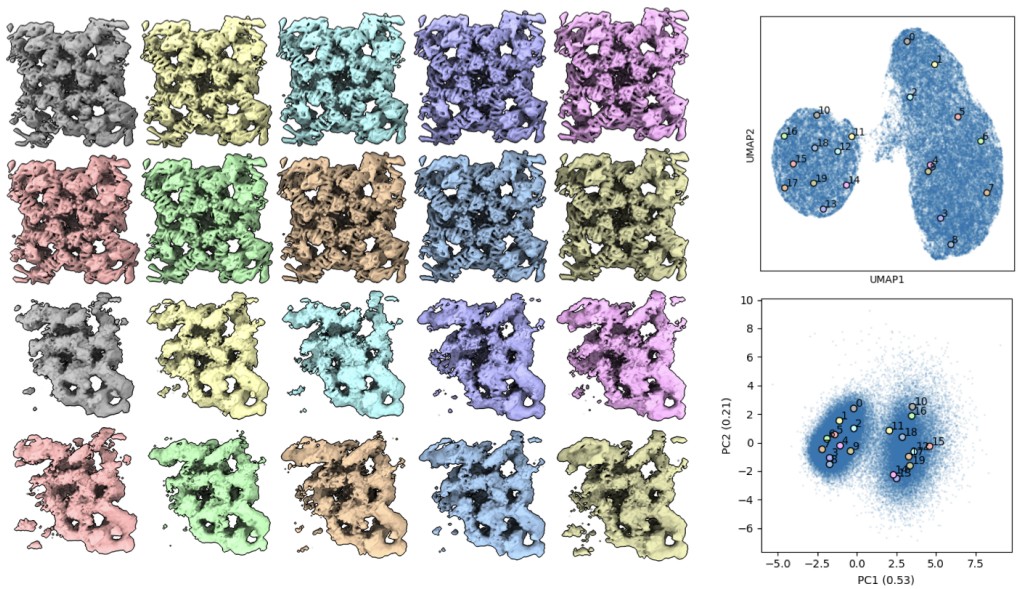

**Fig. S5: Sampling of the DRGN-AI latent space from the ryanodine receptor dataset shows two main clusters corresponding to RyR and non-RyR particles.** *Left:* density maps sampled using k-means clustering with $K = 20$ on the latent space; *right:* latent space plots.

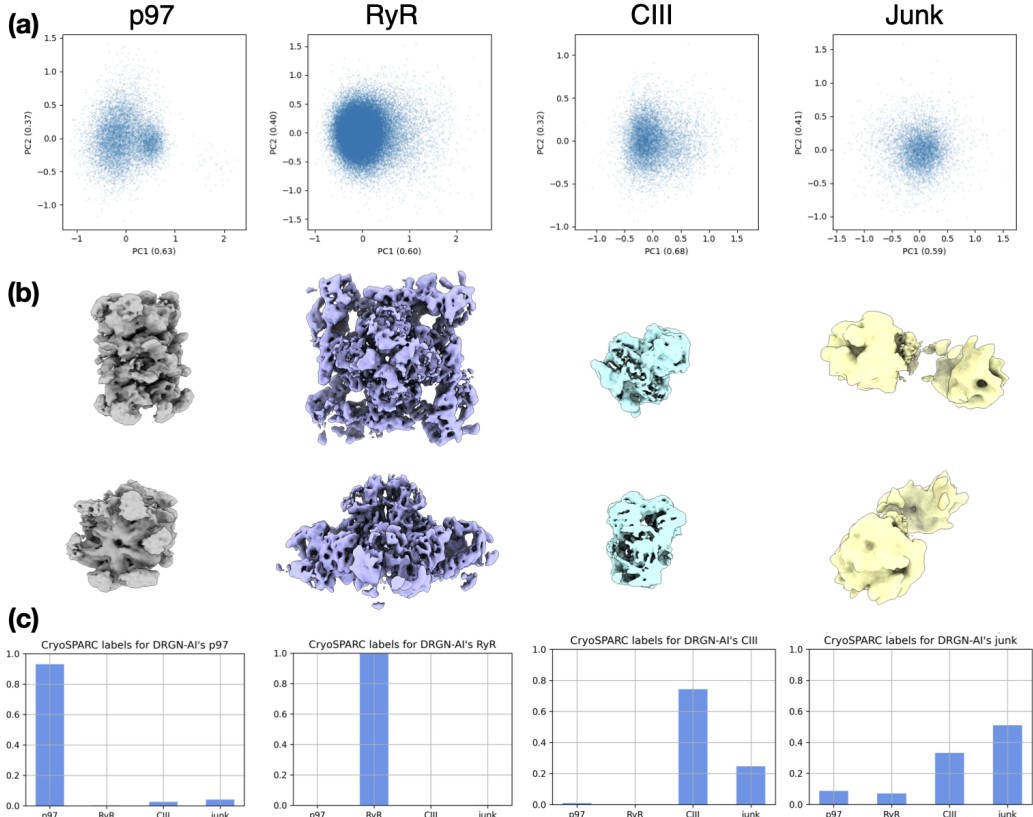

**Fig. S6: Additional qualitative information for Hydra $K = 4$ on experimental ryanodine receptor dataset.**
**(a)** Latent space plots corresponding to each class from Hydra. **(b)** Additional top and side views of each density map generated from Hydra $K = 4$, with a cutaway view of CIII showing resolution of transmembrane helices. **(c)** Bar plots showing agreement between class assignments for Hydra $K = 4$ and cryoSPARC $K = 6$ heterogeneous refinement (particles from the three recovered RyR classes were combined for analysis), normalized by total number of particles in each Hydra class.

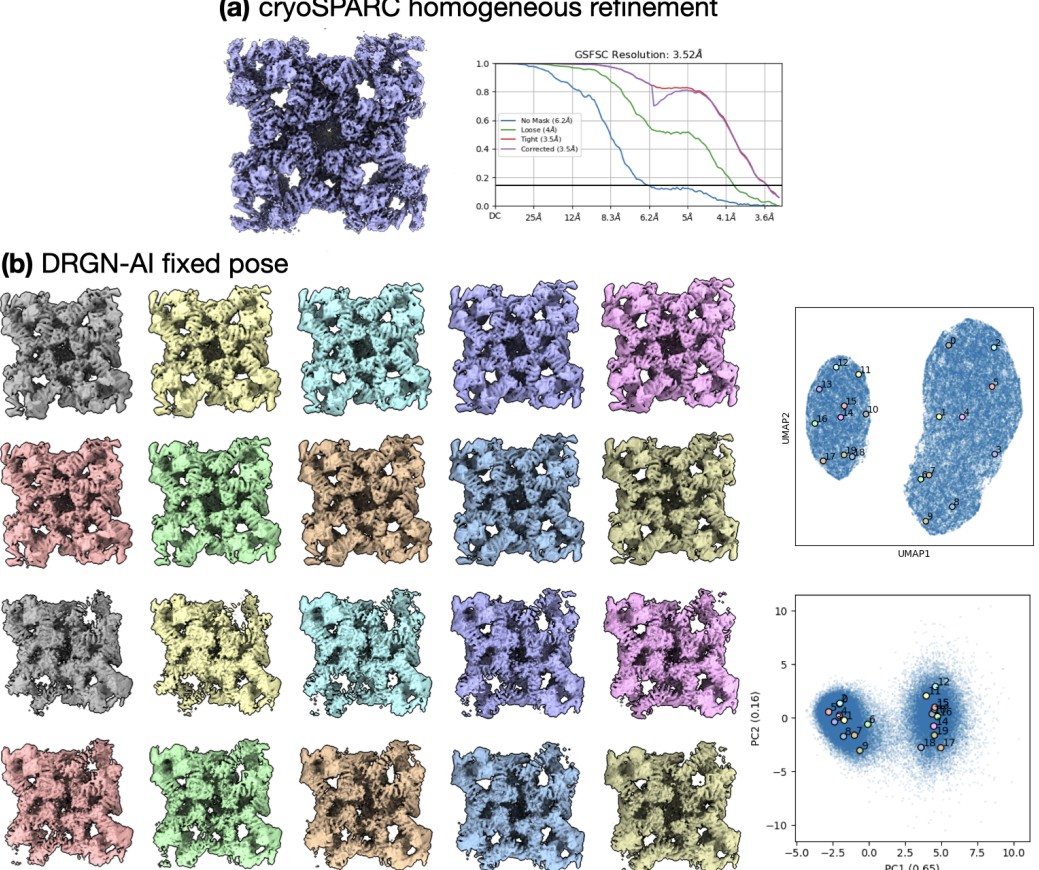

**Fig. S7: The typical processing workflow of generating a consensus reconstruction followed by DRGN-AI heterogeneous reconstruction fails to capture the shape of non-RyR densities, as the cryoSPARC consensus reconstruction conceals compositional heterogeneity and yields a high-resolution density for RyR only.** (**a**) Homogeneous refinement of the entire ryanodine receptor dataset against a cryoSPARC *ab initio* $K = 1$ alignment of the entire dataset (*left*); *right:* FSC curve. (**b**) single-class DRGN-AI fixed pose with poses from the cryoSPARC homogeneous refinement; *left:* densities from k-means 20 sampling of the latent space; *right:* latent space plots.

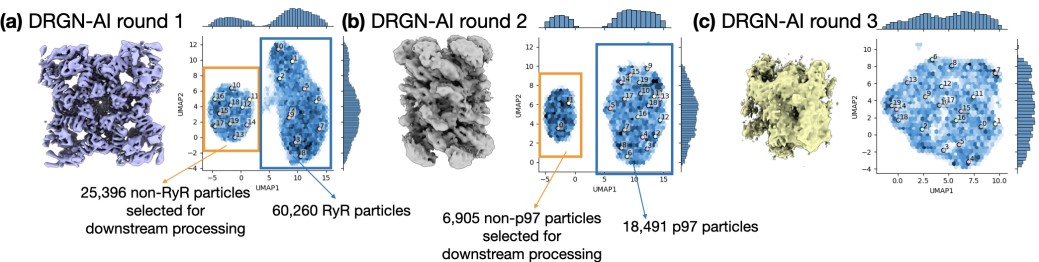

**Fig. S8: Even when using a multi-shot heterogeneous reconstruction approach, DRGN-AI is only able to capture RyR and p97 densities, possibly due to the higher SNR of RyR and p97; the final density shows a mix of CIII and junk particles and no partitioning of the latent space.** (**a**) DRGN-AI on the full experimental ryanodine receptor dataset; resulting RyR density generated from sampling the latent space cluster labeled in blue. (**b**) DRGN-AI on the subset of particles from the latent space cluster labeled in orange from part (**a**); p97 density sampled from latent space cluster labeled in blue. (**c**) DRGN-AI on the subset of particles from the latent space cluster labeled in orange from part (**b**).

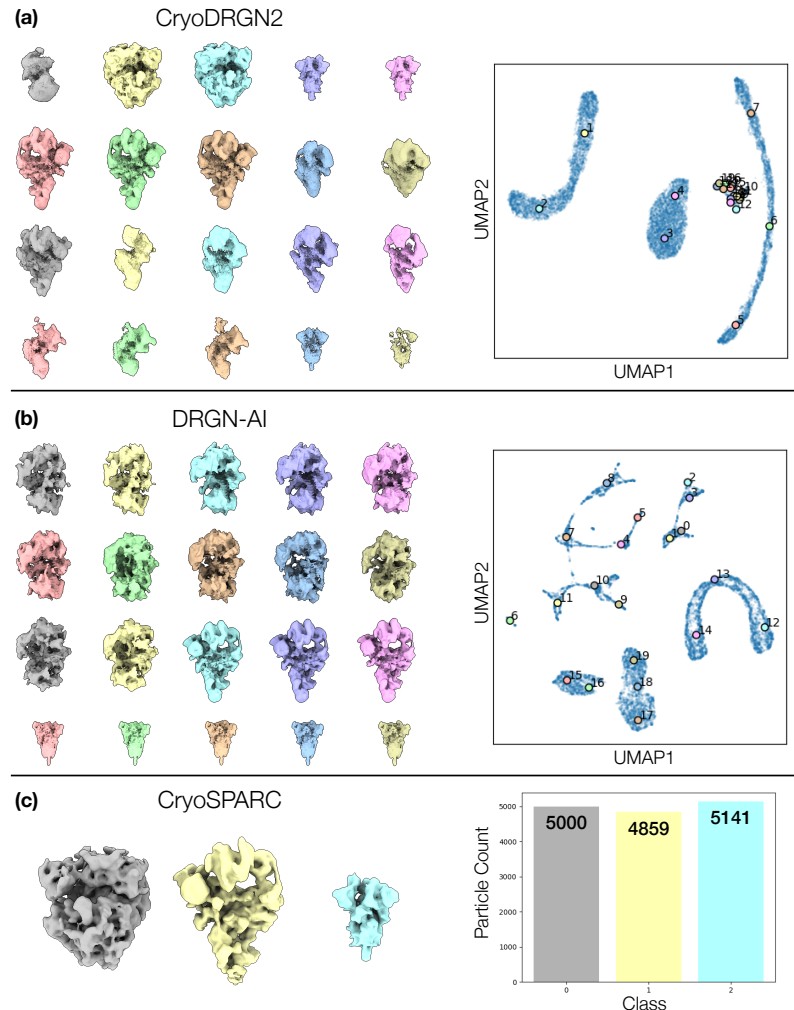

**Fig. S9: Qualitative results of baselines on the synthetic `ribosplike` dataset containing compositional and conformational heterogeneity. (a)-(b)** Conformational space plots and representative K-Means clustering volumes for CryoDRGN2 and single-class DRGN-AI. **(c)** Class volumes and particle counts for CryoSPARC.

# J Additional Tables

| Model | | All | PDB 6up6 | PDB 6id1 | PDB 4cr2 |
|---|---|---|---|---|---|
| | | ARI ↑ | Image-FSC ↑ | Image-FSC ↑ | Image-FSC ↑ |
| Hydra $K=3$ | Replica 1 | **1.00** | $0.24 \pm 0.03$ | $0.393 \pm 0.001$ | $0.364 \pm 0.001$ |
| | Replica 2 | **1.00** | $0.25 \pm 0.03$ | $\underline{0.394 \pm 0.001}$ | $\mathbf{0.367 \pm 0.001}$ |
| | Replica 3 | 0.75 | $0.15 \pm 0.02$ | $0.388 \pm 0.001$ | $\mathbf{0.367 \pm 0.001}$ |
| Hydra $K=5$ | Replica 1 | **1.00** | $0.25 \pm 0.03$ | $\mathbf{0.396 \pm 0.001}$ | $\mathbf{0.367 \pm 0.001}$ |
| | Replica 2 | **1.00** | $\underline{0.27 \pm 0.02}$ | $0.394 \pm 0.001$ | $0.363 \pm 0.001$ |
| | Replica 3 | 0.89 | $\underline{0.18 \pm 0.02}$ | $\mathbf{0.396 \pm 0.001}$ | $0.365 \pm 0.001$ |
| DRGN-AI $K=1$ | Replica 1 | 0.59 | $0.0242 \pm 0.003$ | $0.0396 \pm 0.005$ | $0.042 \pm 0.004$ |
| | Replica 2 | 0.49 | $0.022 \pm 0.002$ | $0.040 \pm 0.004$ | $0.040 \pm 0.004$ |
| | Replica 3 | 0.53 | $0.026 \pm 0.005$ | $0.038 \pm 0.004$ | $0.041 \pm 0.005$ |
| CryoDRGN2 | — | 0.36 | $0.08 \pm 0.01$ | $0.070 \pm 0.006$ | $0.3 \pm 0.1$ |
| CryoSPARC | — | **1.00** | **0.284** | 0.367 | 0.338 |

**Table 3: Extension of Table 1** The classification accuracy is evaluated for each method using the adjusted Rand index (ARI). To evaluate each method's reconstruction quality, we use the mean area under the Fourier shell correlation (FSC) curve for 20 images per class (we report $\pm 1$ standard deviation). We **bold** the best result, and underline the best result for the second best method.

| | Rotation Error ↓ | | | Translation Error ↓ | | | Resolution at 0.5 FSC ↑ | | |
|---|---|---|---|---|---|---|---|---|---|
| $K$ | 6up6 | 6id1 | 4cr2 | 6up6 | 6id1 | 4cr2 | 6up6 | 6id1 | 4cr2 |
| 1 | 122.73 | 80.07 | 1.21 | 4.754 | 19.144 | 0.005 | 44.31 | 82.29 | 9.14 |
| 3 | 114.82 | 1.19 | 1.23 | 2.936 | 0.022 | 0.013 | 9.29 | 9.14 | 9.14 |
| 5 | 123.43 | 1.18 | 1.22 | 2.812 | 0.020 | 0.016 | 9.29 | 9.14 | 9.14 |

**Table 4:** Quantitative results on the `tomotwin` dataset (30k particles) for Hydra with varying $K$. Metrics include median rotational errors (in degrees), median translation errors (in pixels), and resolution in Å at an FSC cutoff of 0.5, where we compare the ground truth volume to a backprojected volume using the predicted poses (Nyquist limit is at 9.00 Å). We note high pose errors for structure `6up6` despite its good reconstruction quality, likely indicative of pose ambiguity induced by the symmetry of the molecule.

| Method | ARI ↑ | Rotation Error ↓ | | | Translation Error ↓ | | | Per-Image AUC ↑ | | |
|---|---|---|---|---|---|---|---|---|---|---|
| | | Splice | Ribo | Spike | Splice | Ribo | Spike | Splice | Ribo | Spike |
| Hydra ($K=3$) | **0.997** | **1.69** | 0.67 | **0.85** | **0.141** | **0.003** | **0.003** | **0.373** | **0.429** | **0.441** |
| DRGN-AI ($K=1$) | 0.994 | 1.87 | 34.18 | 100.09 | 0.155 | 0.494 | 0.023 | 0.353 | 0.064 | 0.206 |
| CryoDRGN2 | 0.986 | 2.06 | **0.59** | 0.99 | 0.611 | 0.006 | 0.011 | 0.357 | 0.424 | 0.416 |
| CryoSPARC | 0.972 | 1.94 | 1.34 | 1.45 | 0.367 | 0.018 | 0.020 | 0.324 | 0.355 | 0.353 |

**Table 5:** Quantitative results on the `ribosplike` dataset. Metrics include particle classification accuracy (Adjusted Rand Index, ARI), median rotational error (in degrees), median translation error (in pixels), and reconstruction quality (per-image area under the FSC curve). Hydra outperforms state-of-the-art methods at jointly capturing conformational and compositional heterogeneity.

