# OpenReview forum: "Mixture of neural fields for heterogeneous reconstruction in cryo-EM"
_NeurIPS.cc/2024/Conference — NeurIPS 2024 poster_

### Official Review · Reviewer_qUkh · 2024-07-03

**Soundness:** 2
**Presentation:** 3
**Contribution:** 1
**Rating:** 4
**Confidence:** 5

**Summary:**

This paper introduces a novel method, Hydra, for ab initio heterogeneous cryo-EM reconstruction. Different from existing approaches, Hydra separately models conformational and compositional heterogeneity by integrating K-parameterized neural fields to represent cryo-EM density maps. Furthermore, Hydra employs a hybrid optimization strategy to optimize particle poses, heterogeneity, and density map representations concurrently. The authors assess the efficacy of Hydra on three datasets comprising various protein complexes (two synthetic, one experimental). Extensive experimental results indicate that Hydra outperforms three baseline methods regarding reconstruction quality, particle classification, and pose estimation accuracy.

**Strengths:**

This paper aims to tackle ab-initio reconstruction – simultaneously estimating poses and reconstructing 3D structure, which is one of the most challenging problems in cryo-EM. The scope of the problem has been further extended to a more challenging setting by assuming the captured particle images exhibit the motions of structure and conformational heterogeneity. Thus, the problem setting is novel and challenging.

This paper proposes Hydra, the first ab-initio heterogeneous cryo-EM reconstruction method based on a mixture of neural networks to estimate conformational and compositional heterogeneity in the training process simultaneously.

**Weaknesses:**

**Synthetic Dataset**

The number of images in the tomotwin3 synthetic dataset is too small to match the real cryo-EM setting; there are only 3000 particle images with a very low SNR of 0.01 (tomotwin3). In a real scenario, there would be 50,000 to more than 100,000 particles with conformational variability. Without reporting the 3D resolutions of the results, it is very hard to quantitatively evaluate the reconstruction results. Additionally, this synthetic dataset lacks conformational variability, which hinders the evaluation of conformational variability recovery. The combination ratio of the three types of particles is not explored either; it remains unclear if Hydra would be sensitive to a class with only a small ratio of particle numbers. In Section 4.3, the dataset settings that include pre-catalytic spliceosome, 80S ribosome, and SARS-CoV-2 spike protein are unrealistic. To sum up, I recognize a significant gap between this synthetic dataset and real datasets, and I feel the experiment is not sufficient to evaluate Hydra adequately.

**Reconstruction Quality Evaluation**

For qualitative evaluation, the differences between the various states in Figure 4 are very subtle, making it difficult to judge whether the surface changes are due to different conformations or the result of applying different thresholds to the density map. Also, I argue that cryoDRGN-AI and cryoDRGN2 also account for conformational heterogeneity without explicitly classifying particles; their qualitative results should also be compared in Figure 4.

For quantitative evaluation in Table 1 and Table 2, the authors only use Img-FSC to compare reconstruction quality. To the best of my knowledge, prior work such as cryoSPARC and cryoDRGN reports widely used 3D resolution calculated by thresholding the FSC curve between two half maps of the reconstruction for experimental datasets or between the reconstructed 3D density map and the synthetic ground truth density map (available for synthetic datasets).

**Choice of Metrics for Pose Evaluation**

The authors use the median Frobenius norm as the metric to compare pose error, which ignores the translation part. Referring to DRGN-AI, the in-plane and out-of-plane angle error, along with the translation error, should be reported. Additionally, showing the angle distribution of poses for comparison could be beneficial.

**Ablation Study**

This paper misses the ablation study of the number of K. I would like to know how to determine K in real cases. In this paper, the authors run CryoSPARC multiple times to determine the best K for H.

**Miscellaneous**

1.	The chirality of the reconstruction results for the pre-catalytic spliceosome in Figure 4 appears to be incorrect.
2.	The construction of the ribosplike dataset involves a mixture of three different proteins. In real single-particle cryo-EM experiments, this scenario seems rare as the purified samples are carefully prepared and should not contain or only contain a very small ratio of undesired particles that can be easily filtered out in 2D classification. Maybe in cryo-ET, Hydra can perform one-for-all sub-tomo averaging?
3.	What’s the protocol to run CryoSPARC on a synthetic dataset? If it performs so well in Table 1, what is the advantage of Hydra?
4.	This paper seems rushed. In Lines 57 – 62, the first three contributions have basically the same meaning; please consider rephrasing them.

**Questions:**

My major concerns have been listed in the weakness section in this review, along with some of my confusion and questions. I list the important ones:
- Why is the synthetic dataset limited to 3000  particle images in a very low SNR of 0.01 for the tomotwin3 dataset? In my experience, this limited number usually causes poor reconstruction results.
- Can you provide the 3D resolutions of the reconstruction results to facilitate a more quantitative assessment?
- Have you tested how Hydra handles classes with a small ratio of particle numbers, given that the combination ratio of the three types of particles is not explored?
- The differences between the various states in Figure 4 are very subtle. How do you ensure that the observed surface changes are due to different conformations and not due to varying density map thresholds?
- Could you include qualitative comparisons with cryoDRGN-AI and cryoDRGN2, as these methods also address conformational heterogeneity without explicit particle classification?
- How do you sample the different conformation states in the conformational latent space? (like the white circular points in the latent space in Figure 4.)
-The median Frobenius norm metric ignores translation errors. Could you also report the in-plane and out-of-plane angle errors, along with the translation error, similar to DRGN-AI?
- What is the protocol for running CryoSPARC on a synthetic dataset, and what advantages does Hydra offer if CryoSPARC performs well in Table 1?
- How is the runtime efficiency of the Hydra algorithm compared to other algorithms?

**Limitations:**

The authors have addressed the limitations of Hydra in the main paper.

---

> ### Author Rebuttal · Authors · 2024-08-07
>
> We thank our reviewer for their comments on the significance and the difficulty of the problem our method addresses. We hope to address their concerns in the following response.
>
> **Synthetic Datasets**
>
> To further validate Hydra on the tomotwin dataset, we generated a larger version of this dataset (30,000 particles) with the same SNR. Results are shown in Figure A2 (see document attached to the shared rebuttal).
>
> For synthetic data like tomotwin, we use the ground-truth maps to quantitatively evaluate the reconstruction results. Following our reviewer’s suggestions, we provide 3D resolutions for the larger tomotwin dataset (30,000 particles) in Table A1 of the attached document.
>
> The tomotwin and ribosplike datasets are built with a uniform distribution over the three possible states, while the three states in the experimental dataset (RyR) are present in very different proportions (RyR: 71%, p97: 8%, CIII: 13%, junk: 8%, according to cryoSPARC and Hydra, Fig 3.b). We demonstrate Hydra’s ability to process all these datasets, confirming the possibility to handle different types of distributions over classes.
>
> We acknowledge the existence of a gap between the SNR in synthetic datasets vs. experimental datasets. However, for synthetic datasets, we compare Hydra to baselines (cryoSPARC, cryoDRGN2, DRGN-AI) using the same level of noise and show an improvement in reconstruction quality (Table 1, Table 2, Fig S7). Our experiments on the real dataset validate Hydra’s ability to handle realistic noise levels and to outperform existing methods (Fig 3.b, S2, S6).
>
> **Reconstruction Quality Evaluation**
>
> - We will include movies of the states shown in Fig 4 to better show their conformational changes.
>
> - Per-image FSC measures per-particle 3D resolution on synthetic datasets (where ground truth maps are available) and is a standard metric for assessing the reconstruction quality of methods that can represent conformational (continuous) heterogeneity. We refer, for example, to Fig 3 and Supplementary Figure 1 of the cryoDRGN paper [1]. For the tomotwin dataset, which only contains compositional heterogeneity, we provide per-class 3D resolutions with a threshold at 0.5 FSC in Table A1 (see document attached to the shared rebuttal).
>
> [1] Zhong, E. D., Bepler, T., Berger, B., & Davis, J. H. (2021). CryoDRGN: reconstruction of heterogeneous cryo-EM structures using neural networks. Nature methods, 18(2), 176-185.
>
> **Choice of Metrics for Pose Evaluation**
>
> In order to better assess pose accuracy, we provide per-class angular and translation errors for the tomotwin dataset in Table A1, and for the ribosplike dataset in Table A2 (see document attached to the shared rebuttal).
>
>
> **Ablation Study**
>
> Fig 2 provides an analysis on the influence of $K$. When $K$ is too low (Fig 2.a), the reconstruction is inaccurate. When $K$ is too large, some of the capacity of the model is not used, but the reconstructed states are accurate (Fig 2.b). To further stress-test Hydra, we provide additional results using $K=7$ on the tomotwin dataset in Fig A1 (see document attached to the shared rebuttal).
>
> To obtain the optimal value of $K$, one can run Hydra several times with increasing values for $K$. Although choosing a larger-than-optimal value for $K$ would lead to unnecessary computation, the quality of the reconstruction does not degrade when $K$ is too large, as shown in Fig 2 (b vs. c).
>
> We acknowledge that this sweeping procedure requires time, energy and memory and could be avoided with a principled way of choosing $K$. This limitation is currently mentioned in the Discussion section (L311-317), and potential mitigation strategies are suggested.
>
> **Miscellaneous**
>
> - We thank our reviewer for pointing out the wrong chirality of the spliceosome in Fig 4. Since the chirality of a molecule is not identifiable from cryo-EM projections, we showed the chirality that we got out of Hydra without modification. We will flip (mirror) the density map to show the correct chirality.
>
> - While the choice of proteins in the ribosplike dataset may be unrealistic, having a mixture of different particle types in cryo-EM is not unrealistic, as evidenced by imaging of a lysate sample in the RyR experimental dataset. We agree with our reviewer: adapting this method to sub-tomogram averaging of _in situ_ mixtures is a promising avenue for future work (L320).
>
> - We always use the default parameters of cryoSPARC, both for synthetic and experimental datasets. We will mention this in the supplements.
>
> - Table 1 focuses on the tomotwin dataset, a synthetic dataset with strong compositional heterogeneity. The Table shows previous neural-based methods fail on this dataset. CryoSPARC performs well, but is unable to reconstruct conformational heterogeneity, which is quantitatively shown in Table 2.
>
> - We will clarify the contributions in L57-62.
>
> **Questions**
>
> - We thank our reviewer for suggesting to provide access to the reconstructed maps. We will follow this suggestion and add the reconstructed maps to the supplementary material, in the _mrc_ format. To make our method fully reproducible, we will provide access to the code upon publication.
>
> - We include qualitative comparisons to DRGN-AI and cryoDRGN2 in Fig S2, S4.b, S6 and S7.
>
> - In Fig 4, the conformational states are manually selected in the latent space. In other figures, we use k-means clustering (L247, L940, caption of Fig S2, S4, S7).
>
> - We evaluate the efficiency of Hydra by comparing GPU runtimes on the tomotwin dataset using a single NVIDIA A100 GPU. CryoDRGN2 required 1h50min. DRGN-AI required 1h20min. Hydra required 4h when $K=3$ and 6h20min when $K=5$. We will include this information on runtime efficiency in the supplemental.

---

> > ### Comment · Reviewer_qUkh · 2024-08-09
> >
> > I sincerely appreciate the authors' huge efforts in responding to the review. I have also carefully considered the perspectives provided by other reviewers. While some of my concerns, such as the choice of evaluation metrics and the size of the synthetic dataset, have been addressed, I still find the overall quality of the paper to fall below the standard expected for NeurIPS. The main technical contributions remain unclear, and the experimental setup is lacking, particularly due to the absence of a real dataset and the low-quality reconstruction results, which are insufficient for 3D model building. Consequently, I am unable to raise my rating at this time. I recommend that the authors extend this work to cryo-ET sub-tomogram averaging, where the motivation is stronger and more substantial real-world experiments can be performed.

---

> > > ### Author Response · Authors · 2024-08-10
> > > **Clarification to response -- demonstration on real data**
> > >
> > > Thank you for reading our response. We realize there was a misunderstanding in your comment, and we apologize for the lack of clarity.
> > >
> > > In our work, we consider two synthetic datasets for validation of our method: tomotwin and ribosplike. The first contains a mixture of 3 complexes that we use for evaluation of hyperparameters (which we updated during rebuttals to contain a realistic number of images, and with resolution metrics); the latter contains a mixture of 3 complexes with conformational motions. However, **we do additionally showcase the method on a real dataset**, collected from a lysate sample where we are able recover the structures of the membrane protein ryanodine receptor (RyR), the mitochondrial respiratory chain complex III (CIII), and the dimeric complex of the soluble valosin-containing protein (p97). We don't emphasize it since there is no ground truth to validate the conformations, however we feel that this dataset demonstrates that the method will be transferrable to real application settings.
> > >
> > > To answer your comment on the technical contributions, we would like to emphasize that Hydra is the first method to use a mixture of neural fields for cryo-EM reconstruction. We show that it can handle a novel and complex problem setting in cryo-EM reconstruction: the simultaneous estimation of conformational and strong compositional heterogeneity, in an _ab initio_ setting.
> > >
> > > We also want to clarify that the new resolution estimates for the volumes are close to the Nyquist resolution for this dataset (9.2 A vs 9 A). We emphasize that these experiments were performed with 128x128 images for computational considerations for hyperparameter evaluation.
> > >
> > > Finally, thank you for the suggestions to extend this work to cryo-ET sub-tomogram averaging (STA). We completely agree this would be an amazing showcase and hope that this work can inspire research in this direction, but we feel that adapting to a new imaging modality and data type is beyond the scope of this work.

---

> > > > ### Comment · Reviewer_qUkh · 2024-08-10
> > > >
> > > > Thank you for the clarification! It might be easier to follow if the order of the results in Section 4.2 and Section 4.3 is reversed.
> > > >
> > > > As the authors mentioned, in an experimental dataset, methods like DRGN-AI and cryoSPARC struggle to reveal the subtleties of the total distribution of compositional heterogeneity without expert data processing. This is the most convincing case, in my opinion, and should be further emphasized in the paper, with additional evaluations based on it. I recommend that Hydra be carefully compared with the most competitive baselines, cryoSPARC and DRGN-AI, in terms of performance, training and inference speed, and the level of expert effort required.
> > > >
> > > > While my concerns about the real dataset have been somewhat addressed, the paper still has some previously mentioned weaknesses that I cannot ignore.I will consider improving my rating before the discussion deadline, taking into account all the comments from the other reviewers.

---

> > > > > ### Author Response · Authors · 2024-08-12
> > > > >
> > > > > Thank you for your suggestion regarding the order of Section 4.2 and 4.3. We will swap these sections for clarity. On the experimental dataset, we have provided a comparison to cryoSPARC and DRGN-AI in Figure S5 and S6, and we agree with the reviewer that this comparison could be emphasized. We will move parts of Fig. S5 and S6 to the main paper and can further add a discussion on the level of expert processing required in the related works and the discussion of the results. Thank you for the feedback!

---

### Official Review · Reviewer_14wr · 2024-07-09

**Soundness:** 3
**Presentation:** 3
**Contribution:** 2
**Rating:** 4
**Confidence:** 4

**Summary:**

In this paper, author(s) porpose Hydra, an *ab initio* approach to model conformational and compositional heterogeneity. They ahieve this by parameterizing structures of proteins as a mixture of *K* neural fields.

**Strengths:**

Originality:

- Authors propose to incorporate neural network ensemble with recent *ab initio* reconstruction method to enhance the ability to model complex heterogeneity.

Quality:

- Through qualitative experiments, authors demonstrate the method is capable of clearly distinguish different compositional states in the synthetic dataset.
- Hydra is able to identify both compositional and conformational heterogeneity in real datasets.
- The method exhibits better quantitative results compared to existing *ab initio* reconstruction methods.

**Weaknesses:**

Novelty:

- The hierarchical pose search (HPS) method for pose estimation is proposed in DRGN-AI [1].
- Using an ensemble of representations to model heterogeneity in protein cryo-EM reconstruction has been adopted in many previous works [2][3].

[1] Levy, Axel, et al. "Revealing biomolecular structure and motion with neural ab initio cryo-EM reconstruction." *bioRxiv* (2024): 2024-05.
[2] Punjani, Ali, and David J. Fleet. "3D variability analysis: Resolving continuous flexibility and discrete heterogeneity from single particle cryo-EM." *Journal of structural biology* 213.2 (2021): 107702.
[3] Kimanius, Dari, Kiarash Jamali, and Sjors Scheres. "Sparse Fourier backpropagation in cryo-EM reconstruction." *Advances in Neural Information Processing Systems* 35 (2022): 12395-12408.

**Questions:**

How would Hydra perform on EMPIAR-10076, one widely used dataset with complex compositional heterogeneity?

**Limitations:**

Authors discussed the limitation in their paper.

---

> ### Author Rebuttal · Authors · 2024-08-07
>
> We thank our reviewer for their comments. We hope to address them in the following response.
>
> **Distinction with Previous Works**
>
> Hydra uses the pose search strategy and autodecoding framework from DRGN-AI [1]. It primarily differs by using several neural networks, “latent scores” (L197) and a new loss function involving a marginalization over the possible classes (Eq 7). Compared to DRGN-AI, Hydra broadens the scope of datasets that can be handled. We will clarify that the pose estimation strategy was already used in DRGN-AI (it is only adapted to consider the simultaneous representation of several classes).
>
> We thank our reviewer for pointing out the relevant reference [3], which we will cite. Although the references [2] (3DVA) and [3] reconstruct a combination of 3D arrays, those are not density maps but rather vectors with the same dimension as density maps, representing the basis of a low-dimensional linear space. The conformation heterogeneity is then represented in the linear space spanned by these vectors (the “structural basis” [3]). Unlike Hydra, 3DVA (cited on L34 and L76) does not handle compositional heterogeneity (L38).
>
> **EMPIAR-10076**
>
> We thank our reviewer for this suggestion. EMPIAR-10076 (assembling ribosome) would indeed be a relevant dataset to demonstrate our method. We will run additional experiments on this dataset and try to include results in the supplement.

---

> > ### Comment · Reviewer_14wr · 2024-08-11
> >
> > I sincerely appreciate authors' clarification. But based on my understanding, 3DVA can handle compositional heterogeneity. In the fig. 9 of the bioRxiv version of 3DVA paper, they showed results on EMPIAR-10076, a dataset with complex compositional heterogeneity.

---

> > > ### Author Response · Authors · 2024-08-12
> > >
> > > Apologies we mis-stated that 3DVA does not handle compositional heterogeneity in our last response – we meant to clarify that 3DVA is not designed to handle a mixture of different species, a strong form of compositional heterogeneity. While 3DVA can technically handle compositional heterogeneity (as any density-based method does not explicitly constrain the reconstructed density maps), 3DVA uses a linear subspace of the volumes for its model of heterogeneity, and thus has an inductive bias towards modeling continuous motions of a single complex (similar to cryoDRGN and DRGN-AI). We would like to emphasize that, in Fig. 9 of 3DVA [1], the reconstructed complex is an assembling 50S ribosome where the discrete “classes” share most of their density. Finally, but perhaps most relevant, 3DVA is applied in the fixed pose setting, typically at the end of the processing pipeline after several steps of 2D or 3D classification, where poses are typically aligned to a single static reference structure, which would not apply in the case of a mixture of proteins.
> > >
> > > In contrast, Hydra is the first method to adopt an ensemble of independent neural representations to model heterogeneity. This approach is similar to the ubiquitous 3D classification (a discrete mixture model of voxel arrays), but is far more expressive due to its neural representation augmented with latent scores. Thus, we demonstrate Hydra on datasets containing a mixture of different species that have conformational motions, where _ab initio_ pose estimation is necessary.
> > >
> > >
> > > [1] Punjani, A., & Fleet, D. J. (2021). 3D variability analysis: Resolving continuous flexibility and discrete heterogeneity from single particle cryo-EM. Journal of structural biology, 213(2), 107702.

---

### Official Review · Reviewer_SenM · 2024-07-10

**Soundness:** 3
**Presentation:** 4
**Contribution:** 4
**Rating:** 7
**Confidence:** 5

**Summary:**

This work describes a new method for ab initio heterogeneous reconstruction in cryo-EM using mixtures of neural fields. This generalizes previous approaches, such as CryoDRGN and DRGN-AI, which attempts to reconstruct 3D molecular densities using a single neural field representation. The resulting method is able to handle both compositional (discrete) and conformational (continuous) heterogeneity, with each mixture component handling the continuous variability of each distinct compositional state. The performance of the method is evaluated on two synthetic and one experimental dataset.

**Strengths:**

This represents a natural and well-structured extension of previous neural-field approaches to cryo-EM reconstruction. The method is well-motivated and described with an appropriate level of detail. Finally, the numerical results verify many important aspects of the proposed method. Overall, the writing is clear and easy to understand.

**Weaknesses:**

The most important issue is the lack of experimental validation for the combined estimation of compositional and conformational heterogeneity. While this is tested in the third experiment (Section 4.3), this is only on a synthetic dataset. As the authors are no doubt aware, however, the behavior of a reconstruction algorithm can be quite different when applied to real data. It is therefore encouraging that the authors present results on an experimental dataset (Section 4.2), but this only covers compositional heterogeneity (and not conformational). That being said, validating the full method on an experimental dataset would make a stronger case for the proposed work.

**Questions:**

– On line 49, please explain the principal difference between DRGN-AI and its predecessors. This is particularly important since the proposed method is quite closely related.
– What is meant by “tackle the discrete heterogeneity and the continuous variability *in sequential order*”? Is this referring to clustering the data, estimating the pose, and then estimating the continuous variability for each cluster? This could perhaps be clarified.
– Another approach to manifold embedding for continuous heterogeneity (line 82) is discussed in Moscovich et al., 2020.
– Line 130 should have “used” instead of “use”.
– The last line in the caption for Figure 1 is missing “Section” in front of “3.3”.
– If the dimension d is the same for each component in the mixture (eq. 3), how does this work when the number of degrees of freedom varies with each discrete structure?
– In eq. 4, σ_2 should be σ^2.
– Please discuss the choice of using point estimates for the pose and latent variables in eq. 6. Why is this approximation reasonable? This can be especially sensitive in the ab initio case where the can be great uncertainty in the pose estimation stage. Why do we not need to account for this here?
– Why use the area under FSC to evaluate the accuracy of the reconstruction (Table 1)? What are the advantages compared to gold-standard resolution estimates?
– Does the “strategy” on line 216 refer to the pose (eq. 9) estimation or to the overall estimation of pose and other variables? In other words, do we run ~1e6 iterations of pose estimation of ~1e6 iterations of pose estimation alternated with SGD?
– In line 227, what is meant by “a single pass”? The described algorithm seems to loop over the data several times over. Single-pass approaches usually refers to moment-based methods which pass over the data once to calculate certain statistics and then use those statistics to reconstruct the molecule.
– What is the SNR of the images described in Section 4.3? This information is important enough that it should be supplied in the main text.
– Please provide running times for the various experiments. The metric of 4 GPU-days is cited in the Discussion, but it is not clear how this relates to the number of images (is this for the 60 000-image dataset or the 3 000-image dataset?).

**Limitations:**

As stated above, the main limitation of the work is its lack of validation on experimental data (for both compositional and conformational variability). It is also not clear how computationally intensive the implementation is and how this can be mitigated.

---

> ### Author Rebuttal · Authors · 2024-08-07
>
> We thank our reviewer for their constructive comments and overall positive rating of our submission.
>
> **Validation on an Experimental Dataset Combining Compositional and Conformational Heterogeneity**
>
> As mentioned by our reviewer, processing real cryo-EM data often comes with unforeseen and significant challenges. By demonstrating Hydra’s ability to process the “ryanodine receptor” (RyR) mixture dataset, we hope to provide evidence that our method can process experimental data, in spite of the challenges and non-idealities that come with it. We agree with our reviewer on the fact that demonstrating the unique capabilities of Hydra on an experimental dataset with mixed heterogeneity, and potentially revealing motion or complexes that could not be seen before, would constitute an exceptional scientific leap. By sharing our work (and our code), we hope to provide new capabilities to the cryo-EM community and enable it to make this leap.
>
> **Questions**
>
> - We will clarify the main differences between Hydra and DRGN-AI in L49: use of several neural networks, the presence of “latent scores” and a new loss function involving a marginalization over the possible classes.
>
> - We will clarify the term “sequential order” on L79. The explanation given by our reviewer is correct.
>
> - We will add a reference to Moscovich et al (Cryo-EM reconstruction of continuous heterogeneity by Laplacian spectral volumes, Inverse Problems, 2020) on L82. Thank you for pointing this out.
>
> - We will add the missing character on L130.
>
> - We will add “Section” in the caption of Fig 1.
>
> - In the current architecture, the conformational latent vectors are stored in a $N$-by-$d$-by-$K$ array, meaning that all classes have the same latent dimension. We hypothesize that the latent dimension $d$ must be greater or equal to the largest number of degrees of freedom among all compositional states. However, it would be possible to use per-class dimensions $d_k$, for example using a dictionary of $K$ arrays of dimensions $N$-by-$d_k$. We will mention this in the Discussion section.
>
> - We will fix the typo in Eq 4, thank you for pointing it out.
>
> - For poses, the point estimates are only used in the second phase of optimization (the Stochastic Gradient Descent phase). During the first phase (Hierarchical Pose Search), poses are exhaustively searched over using the strategy described in Section B (supplementary material). This HPS phase enables Hydra to perform _ab initio_ reconstruction.
>
> - The exhaustive search (Eq 9) is only applied to poses on a predefined number of images (~1e6). The schedule describing the switch from HPS to SGD is further described in L869-871 (supplementary material).
>
> - Following our reviewer’s suggestion, we report per-class resolutions at an FSC cutoff of 0.5 for the tomotwin dataset in Table A1 (see document attached to shared rebuttal). We note that a per-image FSC metric is better suited for the ribosplike dataset as it accounts for conformational differences in the true structural distribution.
>
> - We recognize that the term “single pass” is misleading in L227. We will replace it with “single run”.
>
> - The SNR for the dataset described in Section 4.3 is 0.1. We will add this information to the supplementary information, where the generation protocol is described (Section F).
>
> - The mention of “4 GPU-days” relates to the experimental dataset (85k images). The 3,000 image dataset tomotwin3 was processed using a single NVIDIA A100 GPU and required 4h when $K=3$ and 6h20min when $K=5$. DRGN-AI required 1h20min. We will clarify this point in the supplement.

---

> > ### Comment · Reviewer_SenM · 2024-08-13
> >
> > Thank you for your thorough rebuttal and for your clarifications. I will keep my score at 7 as I believe this is a good paper that deserves to be published as part of the proceedings.

---

### Official Review · Reviewer_RoiP · 2024-07-12

**Soundness:** 3
**Presentation:** 2
**Contribution:** 3
**Rating:** 7
**Confidence:** 3

**Summary:**

The paper presents a neural network-based methodology for modeling both compositional and conformational protein states in cryo-electron microscopy (Cryo-EM) 3D reconstruction.

In particular, the authors propose a fully *ab initio* approach, named Hydra, which enables the joint inference of poses, conformations, and class identities.

The novelty of this approach lies in its ability to capture both discrete (compositional) and continuous (conformational) heterogeneity within Cryo-EM datasets, without relying on pre-computed pose estimations. Previous methods have either struggled with accurate pose estimation, relied on coarse initializations and upstream algorithms, or had limited capacity to represent complex biomolecular mixtures.

Moreover, the authors validate their proposed approach by comparing it against other popular methodologies, using both synthetic and real datasets, showing the potential of Hydra to capture both compositional and conformational heterogeneity.

**Strengths:**

The paper proposes a novel approach that improves over previous methods. In particular, the proposed method extends previous work, DRGN-AI, to use mixture models of multiple neural fields instead of a single neural field to model conformal heterogeneity. In addition, the proposed approach can directly classify the reconstructed sample between 1 of K different classes, providing advantages over methods that rely on downstream classification tasks.  Bibliographic references are exhaustive and well-discussed. The provided results highlight the significance of the method, showing substantial improvements over three pre-existing methods.

**Weaknesses:**

Although the manuscript is generally well-written, it may be hard to follow for someone who is not an expert in the field. In particular, there are some areas where accessibility to a broader readership might be possible:
* For me, reading some previous work was necessary to understand the context of this work sufficiently to appreciate and understand the contributions of this manuscript. A clearer introduction tailored to a broader readership could make this submission more self-contained, which I would find desirable.
* The introduction lacks a straightforward definition of the taxonomy used throughout the paper. While the authors introduce the context of their research, in particular regarding cryo-EM, I think that the reader should be introduced to the concepts of “poses”, “conformational states”, and “compositional states”, and then to why they are relevant to the presented research and future users. Casting this into a concise but clear way will be much appreciated by future readers, I believe. After that, the authors address how their technical contributions addressed the main challenges presented to them. In my opinion, such changes will allow non EM experts to appreciate the presented work much better and potentially allow other fields to benefit from the same/similar ideas and methods.
* The proposed method seems to rely heavily on the DRGN-AI approach from Levy et. al. Although the authors explicitly state in the introduction that this paper represents an extension of that method, throughout the paper it’s not always clear whether the methodological choices described are novel contributions or are unaltered from the previous method. For example, in Section 3.4, the sentence “We use the pose estimation strategy introduced in [25]” may lead the reader to think that the authors are experimenting with a new pose estimation approach from the literature, while it was already used in the DRGN-AI (or at least I believe so).
* While results are clear and easy to follow, they only present standard deviations in Table 1 and not in the other tables. Moreover, it is unclear why the CryoSPARC result in Table 1 does not report any standard deviation.
* Will the authors provide a public code repository enabling others to replicate the presented results and use the method in the context of their data and experimentation?

**Questions:**

- Do you think that the choice of using only 2 dimensions to represent the conformational space in Hydra may have affected the performance of your method? I think that the choice is excellent for increasing the interpretability of results since it eliminates the need to use techniques like PCA to visually understand the learned latent space. My intuition is that thanks to the K neural fields, the conformal representation may be better partitioned in lower dimensional spaces, but I wonder if you also have considered a higher number of dimensions.

- In Table 2, both Img-FSC and Pose errors are reported. It is unclear to me if the pose error and the Image FSC may be somehow correlated, as, intuitively, a wrong estimated pose could increase the reconstruction error. If that was the case, then I would be surprised to see such a small gap in Img-FSC with respect to CryoDRGN2 compared to the improvement in pose error.

- In Table 2, Hydra shows the lowest Pose error, while DRGN-AI shows the highest despite using a similar pose estimation approach. Can this performance difference be ascribed to the mixture model, or there are other factors to take into account, e.g. the dimensionality of the conformational space?

**Limitations:**

- The main limitation of this work is the need to know the number of classes K in advance. This can be addressed by an exhaustive search of the optimal value of K, however, this further worsens the second main limitation that is related to the computational cost of the method. The authors address both limitations and propose a possible future direction to reduce computational cost.

- Another point of view of the previous limitation is related to scalability. Given a fixed computational budget, since a new neural field is required for each additional protein class, the possible choice of the number of desired classes K is limited, potentially reducing the applicability of the method in datasets that contain a higher number of different macromolecules.

---

> ### Author Rebuttal · Authors · 2024-08-07
>
> We thank our reviewer for their constructive feedback and suggestions on ways to improve the clarity of the manuscript.
>
> **Clarification of Prior Work and Contributions**
>
> We thank our reviewer for pointing out the lack of clarity in the presentation of prior works and apologize for the absence of important context. We will review our introduction and make an effort to make it clear for a broader audience.
>
> **Clarification of Field-Specific Terminology**
>
> Again, we apologize for the oversight on using field-specific terminology (pose, conformational/compositional heterogeneity). We will clarify these terms in order to allow non-cryo-EM experts to appreciate the presented work and potentially borrow from its ideas.
>
> **Distinction with DRGN-AI**
>
> Hydra uses the pose search strategy and the autodecoding framework from DRGN-AI. It primarily differs by using several neural networks, “latent scores” (L197) and a new loss function involving a marginalization over the possible classes (Eq 7). By doing so, Hydra broadens the scope of datasets for cryo-EM reconstruction methods. As suggested by our reviewer, we will clarify that we use the pose estimation strategy in DRGN-AI and that it is adapted to cope with the simultaneous representation of multiple maps.
>
> **Standard Deviations in Table 1**
>
> The standard deviations reported in Table 1 characterize the spread of the area under the Fourier shell correlation obtained over 20 images (with different conformational states $z$) per class. This can only be measured for neural-based methods (Hydra, DRGN-AI, cryoDRGN2), hence the absence of standard deviation for cryoSPARC, which outputs a single conformational state per class.
>
> **Code Release**
>
> Yes, we will release our code upon publication.
>
> **Questions**
>
> - We agree with our reviewer’s intuition. The use of $K$ neural networks increases the expressivity of the representation and probably decreases the number of latent dimensions required to capture conformational motion. Empirically, we found that a dimension of two was sufficient to obtain accurate density maps, and to capture continuous motion in the ribosplike dataset (Fig 4). However, if Hydra had to process a dataset where one of the compositional states had strictly more than two degrees of freedom, the dimensions of the latent space would have to be increased. We hypothesize that this dimension must be greater or equal to the largest number of global degrees of freedom among all compositional states. We will add a brief discussion on the choice of the latent dimension in the last section.
>
> - Our reviewer’s intuition on the correlation between pose error and image-FSC is correct. In order to provide a clearer assessment of each method’s performance, we show updated metrics for the ribosplike dataset, including rotation and translation errors, in Table A2 (see document attached to the shared rebuttal).
>
> - To show that the poor performance of DRGN-AI on the ribosplike dataset is not linked to the low dimension of the latent space, we give eight dimensions to the latent space of DRGN-AI (L244). We therefore hypothesize that Hydra’s ability to reconstruct this dataset can be ascribed to the mixture model. We provide qualitative reconstructions and a UMAP plot of the conformations obtained with DRGN-AI in Fig S7.

---

### Official Review · Reviewer_u3f9 · 2024-07-13

**Soundness:** 2
**Presentation:** 2
**Contribution:** 2
**Rating:** 4
**Confidence:** 4

**Summary:**

This submission presents Hydra, a method for handling heterogeneous cryo-EM reconstruction. Hydra can model both conformational and compositional heterogeneity and can perform ab initio reconstruction. To achieve this, it parameterizes structures as arising from one of K neural fields. In the optimization pipeline, the conformations, poses, class probabilities, and neural fields are optimized to maximize the likelihood of the observed images. The authors demonstrate the reconstruction of multiple protein complexes from an experimental dataset.

**Strengths:**

1. The proposed approach can handle ab initio reconstruction, meaning it does not require pre-computed image poses.
2. The capability of handling compositional heterogeneity is not well-explored.
3. Results on experimental datasets are provided, with comparisons to baseline methods.

**Weaknesses:**

1. Compared to DRGN-AI, the proposed approach primarily changes the single neural representation to multiple ones.
2. The determination of K seems tricky.
3. The paper lacks comparisons with conventional approaches such as cryoSPARC and RELION, especially qualitative comparisons. More results, preferably video results, are needed.
4. It is unclear whether the pose predictions are accurate. More visualization analysis would be helpful.

**Questions:**

1. How should the value of K be determined? How would the proposed model behave if K is too large?
2. Will the multiple neural representation design lead to excessive partitioning, that is, splitting complete independent structures into unreasonable parts?
3. How does the proposed approach compare to conventional methods like cryoSPARC and RELION in terms of reconstruction quality and efficiency?

---

> ### Author Rebuttal · Authors · 2024-08-07
>
> We thank our reviewer for their comments. We hope to address their questions and concerns in the following response.
>
> **Distinction with DRGN-AI**
>
> Hydra uses the same pose search strategy and autodecoding framework as DRGN-AI. It primarily differs by using several neural networks, “latent scores” (L197) and a new loss function involving a marginalization over the possible classes (Eq 7). By doing so, Hydra broadens the scope of datasets for cryo-EM reconstruction. In particular, we demonstrate that, unlike Hydra, DRGN-AI fails at processing datasets with strong compositional heterogeneity, both on synthetic (Fig 2.a, S7.b) and experimental (Fig 3.d, S2, S6) datasets.
>
> We will clarify, in the description of the method, what elements are borrowed from DRGN-AI.
>
> **Determination of $K$**
>
> The optimal value of $K$ can be obtained by running Hydra several times with increasing values for $K$. Although choosing a larger-than-optimal value for $K$ would lead to unnecessary computation, we find that the quality of the reconstruction does not degrade when $K$ is too large, as shown in Fig 2 (b vs. c). We provide additional results using $K=7$ on the tomotwin dataset in Fig A1 (see document attached to the shared rebuttal).
>
> We acknowledge that this sweeping procedure requires time, energy and memory and could be avoided with a principled way of choosing $K$. However, other conventional approaches for heterogeneous reconstruction (cryoSPARC, RELION) are limited in the same way, while not being able to handle conformational heterogeneity (Fig S7.c). This limitation is currently mentioned in the Discussion section (L311-317), and potential mitigation strategies are suggested.
>
> **Comparison to Conventional Approaches**
>
> Hydra is qualitatively compared to cryoSPARC on the experimental dataset (Fig S5) and on the synthetic ribosplike dataset (Fig S7). We thank our reviewer for suggesting to provide access to the reconstructed maps. We will follow this suggestion and add the maps reconstructed with cryoSPARC, DRGN-AI and Hydra to the supplementary material, in the _mrc_ format.
>
> **Metrics on Pose Estimation**
>
> To better assess the accuracy of pose estimation, we provide both translation and rotation errors for the tomotwin (Table A1) and ribosplike (Table A2) datasets. For rotation errors, we use the geodesic distance in $SO(3)$ (angular error), in degrees, which is a more interpretable metric than the Frobenius norm. We will add these metrics to the supplementary material.

---

> > ### Comment · Reviewer_u3f9 · 2024-08-12
> >
> > Thanks for the author's rebuttal, I have read it carefully as well as other reviewers' opinions.
> >
> > I noticed that the proposed method has been compared with cryoSPARC. What I want to hear is some insights about the advantages of the proposed method compared with these traditional methods, and why structural biologists should use the proposed method instead of sticking to traditional methods?
> >
> > I hope to hear the author's answer during the discussion stage to decide my final rating.

---

> > > ### Author Response · Authors · 2024-08-12
> > >
> > > Thank you for bringing up this discussion point. Our main motivation for Hydra is to propose a method designed to tackle a more challenging class of datasets for cryo-EM – those containing a mixture of distinct, dynamic protein complexes. We note that this is a new experimental setting for cryo-EM, thus we hope that Hydra will inspire new experimental protocols as well as other future reconstruction methods designed for this form of extreme heterogeneity.
> > >
> > > Hydra performs _ab initio_ reconstruction, i.e. joint inference of poses and structure, to address this form of extreme heterogeneity. This contrasts with traditional tools that require input poses (e.g. cryoSPARC 3DVA, cryoSPARC 3DFlex, and most other heterogeneity analysis tools). Since poses are typically obtained from an upstream consensus reconstruction, this assumes that all images can be aligned to a static reference structure, which does not hold when analyzing a mixture of distinct complexes.
> > >
> > > Hydra is the first method to adopt an ensemble of independent neural representations to model heterogeneity. This approach is inspired by the ubiquitous 3D classification (a discrete mixture model of voxel arrays), but is far more expressive due to its neural representation augmented with latent scores. In Fig. 4, for example, we show that Hydra reveals the conformational heterogeneity of the ribosome, the spliceosome and the spike protein, while 3D classification in cryoSPARC can only reconstruct three static states (Fig. S7.c).

---

> > > > ### Comment · Reviewer_u3f9 · 2024-08-13
> > > >
> > > > Thanks for the response. I will consider it for my final decision.

---

### Author Rebuttal · Authors · 2024-08-07

We thank all our reviewers for their detailed and constructive feedback. We value their appreciation of the significance of this work **[RoiP, qUkh]**, its novelty **[14wr]** and the validation of our method on experimental data **[u3f9, 14wr]**. We appreciate them highlighting the substantial improvements demonstrated by Hydra over pre-existing methods **[RoiP, 14wr]** and the relevance of our quantitative **[SenM]** and qualitative **[14wr]** results. We are also glad they emphasized the clarity of the manuscript **[SenM]**.

We provide additional figures and tables in the attached document.

---

### Decision · Program_Chairs · 2024-09-25

**Decision:**

Accept (poster)

**Comment:**

This paper received mixed reviews.  The authors submitted a rebuttal with some additional experimental results and there was an active discussion period with many of the issues noted in the initial reviews being addressed.

The main issues raised were around datasets and novelty.  The issues around datasets were largely resolved and clarified during the discussion phase.  The question of novelty is somewhat subjective with two reviewers (RoiP and SenM) arguing that the work makes a sufficient contribution.  After reviewing the comments and the paper, the AC agrees with RoiP and SenM that the work presents a novel and significant improvement over existing work and therefore recommends acceptance.